# Driver lines for studying associative learning in *Drosophila*

**Yichun Shuai\*, Megan Sammons[†], Gabriella R Sterne[‡], Karen L Hibbard, He Yang, Ching-Po Yang[§], Claire Managan, Igor Siwanowicz, Tzumin Lee[§], Gerald M Rubin, Glenn C Turner, Yoshinori Aso\***

Janelia Research Campus, Howard Hughes Medical Institute, Ashburn, United States

**\*For correspondence:**
shuaiy@janelia.hhmi.org (YS);
asoy@janelia.hhmi.org (YA)

**Present address:** [†]Technion-Israel Institute of Technology, Haifa, Israel; [‡]Department of Biomedical Genetics, University of Rochester Medical Center, Rochester, United States; [§]Life Sciences Institute, University of Michigan, Ann Arbor, United States

**Competing interest:** The authors declare that no competing interests exist.

## eLife Assessment

This **important** collection of over 800 new cell type-specific driver lines will be an invaluable resource for researchers studying associative learning in *Drosophila*. Thoroughly characterized and well documented, this collection will permit researchers to selectively target neurons that deliver information to, or receive it from, the memory center of the fly brain called the Mushroom Body. Given the wealth of new drivers and the genetic access they provide to over 300 cell types, this **compelling** work will be of interest not only to researchers studying the mechanisms of associative learning but more generally to those dissecting sensorimotor circuits in the fly nervous system.

**Abstract** The mushroom body (MB) is the center for associative learning in insects. In *Drosophila*, intersectional split-GAL4 drivers and electron microscopy (EM) connectomes have laid the foundation for precise interrogation of the MB neural circuits. However, investigation of many cell types upstream and downstream of the MB has been hindered due to lack of specific driver lines. Here we describe a new collection of over 800 split-GAL4 and split-LexA drivers that cover approximately 300 cell types, including sugar sensory neurons, putative nociceptive ascending neurons, olfactory and thermo-/hygro-sensory projection neurons, interneurons connected with the MB-extrinsic neurons, and various other cell types. We characterized activation phenotypes for a subset of these lines and identified a sugar sensory neuron line most suitable for reward substitution. Leveraging the thousands of confocal microscopy images associated with the collection, we analyzed neuronal morphological stereotypy and discovered that one set of mushroom body output neurons, MBON08/MBON09, exhibits striking individuality and asymmetry across animals. In conjunction with the EM connectome maps, the driver lines reported here offer a powerful resource for functional dissection of neural circuits for associative learning in adult *Drosophila*.

## Introduction

In the insect brain, the mushroom body (MB) serves as the center for associative learning (***Davis, 2023***; ***Figure 1A-C***; reviewed in ***Davis, 2023***; ***Heisenberg, 2003***; ***Modi et al., 2020***; ***Owald and Waddell, 2015***; ***Rybak and Menzel, 2017***). Information about sensory inputs such as odor and color which can serve as a conditioned stimulus (CS), comes into the calyx of the MB. In *Drosophila*, approximately 2000 Kenyon cells (KCs), the MB's primary intrinsic neurons, represent the identity of sensory stimuli by their sparse activity patterns (***Honegger et al., 2011***; ***Perez-Orive et al., 2002***; ***Turner et al., 2008***). Dopaminergic neurons (DANs) transmit signals related to the unconditioned stimulus (US), such as sugar rewards or electric shock punishments, to the MB (***Burke et al., 2012***; ***Kirkhart and Scott, 2015***; ***Liu et al., 2012***; ***Mao and Davis, 2009***; ***Schwaerzel et al., 2003***). DANs and MB output neurons (MBONs) collectively form 15 compartmental zones that tile down the length of the

KC axons in the MB lobes (*Aso et al., 2014a*; *Tanaka et al., 2008*). Memories are stored as altered weights of synaptic connections between KCs and MB output neurons (MBONs) in each compartment (*Hige et al., 2015a*; *Owald et al., 2015*; *Pai et al., 2013*; *Plaçais et al., 2013*; *Séjourné et al., 2011*). Relative activity levels of MBONs across compartments represent the valence of the learned CS and drive memory-based behaviors (*Aso et al., 2014b*; *Owald et al., 2015*).

The recently completed electron microscopy (EM) connectomes of the *Drosophila* brain in larvae and adults revealed thousands of interneurons upstream of DANs, which convey reinforcement signals to the MB, and downstream of MBONs, which link the MB to premotor pathways and other higher-order brain centers (*Dorkenwald et al., 2023*; *Eichler et al., 2017*; *Eschbach et al., 2020*; *Hulse et al., 2021*; *Li et al., 2020*; *Scheffer et al., 2020*; *Winding et al., 2023*; *Zheng et al., 2018*). Functional investigation of these interneuron cell types has been limited by the lack of cell-type-specific driver lines.

Using the intersectional split-GAL4 method (*Luan et al., 2020*; *Luan et al., 2006*), we previously generated 93 split-GAL4 driver lines that allowed for precise genetic access to 60 MB cell types, including most of the KCs, DANs, MBONs and other modulatory neurons in the MB lobe regions (*Aso et al., 2014a*). These lines have been instrumental in revealing the neural circuit logic by which the MB forms associative memories (*Aso et al., 2019*; *Awata et al., 2019*; *Berry et al., 2018*; *Dolan et al., 2018*; *Felsenberg et al., 2017*; *Handler et al., 2019*; *Hattori et al., 2017*; *Hige et al., 2015b*; *Ichinose et al., 2015*; *König et al., 2019*; *Martinez-Cervantes et al., 2022*; *Masek et al., 2015*; *McCurdy et al., 2021*; *Pavlowsky et al., 2018*; *Plaçais et al., 2017*; *Sayin et al., 2019*; *Shyu et al., 2017*; *Tsao et al., 2018*; *Vogt et al., 2016*; *Wu et al., 2017*; *Yamada et al., 2023*; *Zhang et al., 2019*).

Since the MB split-GAL4 lines were generated, new genetic and computational tools have expanded the cell types that can be targeted and facilitated the split-GAL4 design. Critically, a new collection of ZpGAL4DBD and p65ADZp hemidrivers became available (*Dionne et al., 2018*; *Tirian and Dickson, 2017*). Moreover, the expression patterns of the original GAL4 driver lines were imaged with higher-resolution confocal microscopy and Multi-Color-Flip-Out (MCFO) labeling method to reveal the morphology of individual neurons (*Meissner et al., 2023*; *Nern et al., 2015*). Additionally, advanced tools for computational neuroanatomy were developed to aid the design of split-GAL4 driver lines (*Bogovic et al., 2020*; *Costa et al., 2016*; *Masse et al., 2012*; *Meissner et al., 2023*; *Otsuna et al., 2018*). Using these tools and resources, additional collections of split-GAL4 lines were generated for the atypical MBONS, which have dendritic input both within the MB lobes and in adjacent brain regions (*Rubin and Aso, 2024*). In this report, we introduce a novel collection of approximately 800 split-GAL4 lines, covering sensory neurons for sugar, wind and nociception, projection neurons for olfactory, thermo/hygro-sensory and gustatory signals, ascending neurons from ventral nerve cord (VNC), cell types within the MB, and interneurons that connect with DANs and/or MBONs. While our primary objective was to generate driver lines for studying associative learning, the collection also includes cell types tailored for various other functions. We provide a lookup table (*Supplementary file 1*) that maps the corresponding EM neurons in the hemibrain connectome for these drivers to facilitate connectome-guided investigation of neural circuits. This expanded collection of driver lines will be instrumental for many future studies of associative learning and beyond.

## Results and discussion
### Split-GAL4 design and anatomical screening

We screened the expression patterns of over 4000 intersections of split-GAL4 hemidrivers to identify lines potentially labeling neurons of interest (*Figure 1D*). From this we selected 1183 split-GAL4 lines for further characterization using both higher resolution imaging, and MCFO to visualize the individual neurons that compose each split-GAL4 pattern. For these lines, we employed higher resolution confocal microscopy and visualized the individual neurons that compose each split-GAL4 pattern with the MCFO method. We eventually identified 828 lines that we deemed experimentally useful based on their specificity, intensity and consistency. These fly lines are now publicly available through the webpage of the Janelia Flylight team project (https://splitgal4.janelia.org/cgi-bin/splitgal4.cgi), where we have deposited a total of 28,376 confocal images from 6374 tissue samples to document their expression patterns. We included lines with off-target expression, as they can be valuable for anatomical, developmental or functional imaging experiments, even if not suitable for behavioral

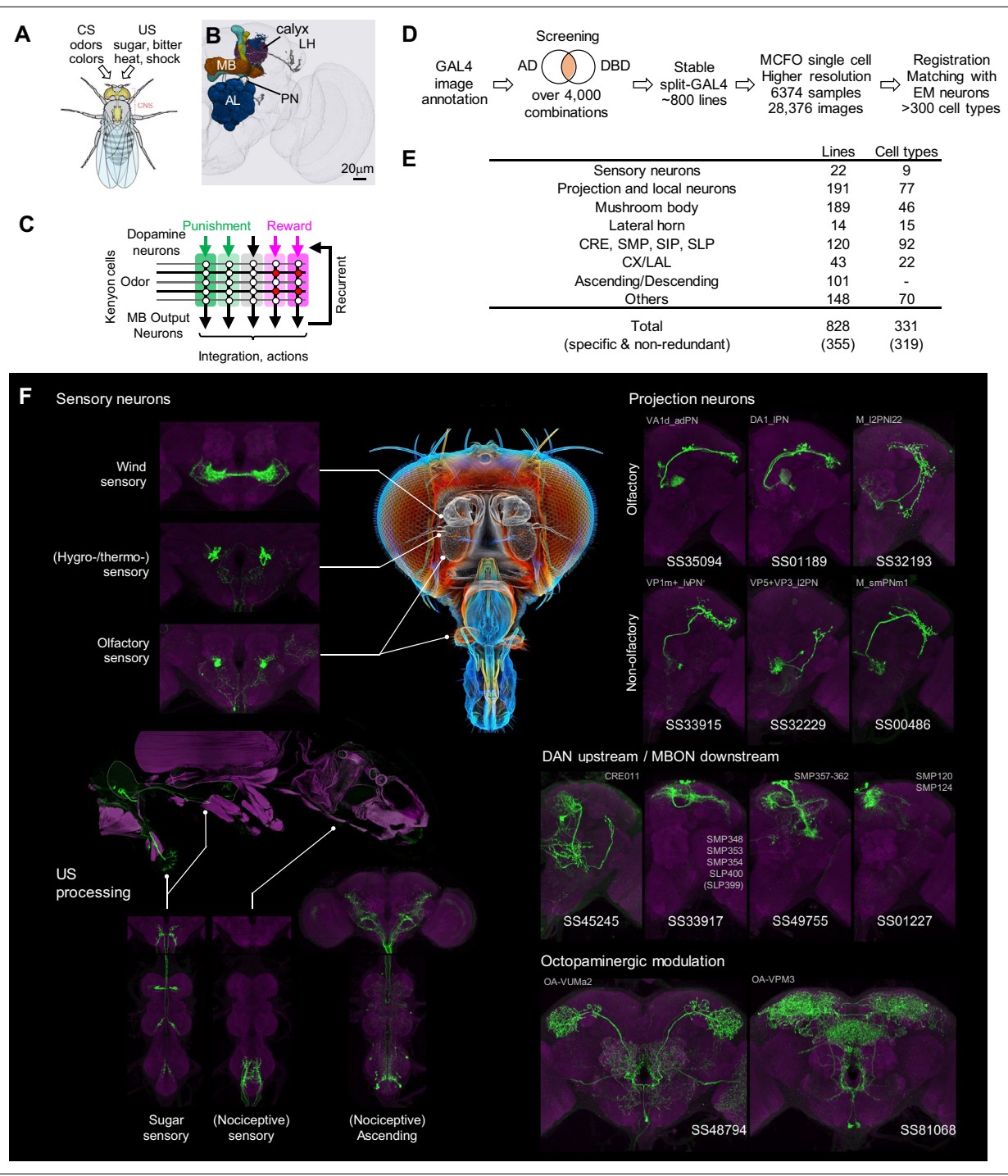

**Figure 1.** Generation and annotation of split-GAL4 lines. (**A**) In associative learning, flies adjust their behavioral responses to conditioned stimuli (CS), such as odors and colors, based on the contingency with innately rewarding or punishing unconditioned stimuli (US), such as sugar, bitter, shock and heat. A schematic of *Drosophila melanogaster* is from **Namiki et al., 2018**. (**B**) An image of a standard fly brain with a rendering of the mushroom bodies (MB) and the antennal lobes (AL). Projection neurons (PN) convey information from the AL to the calyx of the MB and the lateral horn (LH). (**C**) A simplified diagram of the mushroom body circuit. The identity of sensory stimuli is represented by sparse activity patterns of Kenyon cells (KCs). A subset of dopaminergic neurons (DANs) respond to punishment/reward. Dopamine modulates weights of synapses between KCs and MB output neurons (MBONs) depending on the activity status of KCs. The skewed activity patterns of MBONs across compartments in response to the learned stimulus drive memory-based actions and feedback pathways to DANs. (**D**) A summary of the workflow to generate split-GAL4 lines. (**E**) Coverage of the collection. The crepine (CRE), the superior medial protocerebrum (SMP), the superior intermediate protocerebrum (SIP) and the superior lateral protocerebrum (SLP) are MB adjacent brain areas where MBONs and DANs most often have arborizations. CX, central complex. LAL, lateral

Figure 1 continued

accessory lobes. (**F**) Examples of cell types covered by the collection. Expression patterns of CsChrimson-mVenus (green) are shown with neuropil counterstaining of Bruchpilot (Brp) with nc82 antibody (magenta). The whole body image of Gr64f-Gal4 line at the left middle panel is shown with muscle counterstaining (magenta). Light gray labels indicate EM-identified neurons labeled by each line (see *Supplementary file 1* for details). Putative cell types are bracketed.

The online version of this article includes the following figure supplement(s) for figure 1:

**Figure supplement 1.** Examples of cell types covered by the split-GAL4 lines in this collection.

experiments. Additionally, we retained drivers that serendipitously had specific and likely useful labeling of cell types we were not intentionally screening for. Examples of confocal microscopy images are shown in *Figure 1F*, *Figure 1—figure supplement 1*.

We have annotated our split-GAL4 lines by matching the labeled neurons to their counterparts in the hemibrain connectome volume (*Scheffer et al., 2020*). We utilized confocal images registered to a standard brain, and matched neuronal cell types in each split-GAL4 line with those present in other lines and with the EM-reconstructed neurons (*Figure 2A–D*, see Materials and methods). This light microscopy (LM) to EM matching process allows us to locate the cell type of each driver line in the connectome map, enabling users to select driver lines for further functional investigations based on their upstream and downstream synaptic partners (*Figure 2E*; *Figure 2—figure supplements 1–20*).

*Figure 1E* provides an overview of the categories of covered cell types. Among the 828 lines, a subset of 355 lines, collectively labeling at least 319 different cell types, exhibit highly specific and non-redundant expression patterns are likely to be particularly valuable for behavioral experiments. *Supplementary file 1* lists 859 lines (including split-LexA) and their detailed information, such as genotype, expression specificity, matched EM cell type(s), and recommended driver for each cell type. A small subset of 47 lines from this collection have been previously used in studies (*Aso et al., 2023*; *Dolan et al., 2019*; *Gao et al., 2019*; *Scaplen et al., 2021*; *Schretter et al., 2020*; *Takagi et al., 2017*; *Xie et al., 2021*; *Yamada et al., 2023*). The newly generated LexA, Gal4DBD and LexADBD lines are listed in *Supplementary file 2*.

## Drivers for the MB cell types, MBON-downstream and DAN-upstream

Our initial efforts to identify cells of interest started prior to the completion of the EM connectome. At this early stage, we attempted to identify cell types either downstream of MBONs or upstream of DANs using confocal images of GAL4 drivers registered to a standard brain (*Bogovic et al., 2020*). We searched for GAL4 drivers containing cell types with potential connections to MBONs or DANs by quantifying the number of voxels overlapping with MBON axons or DAN dendrites (*Otsuna et al., 2018*). We then built split-GAL4 intersections from selected pairs of drivers from the established hemidriver library (*Dionne et al., 2018*; *Tirian and Dickson, 2017*).

Once EM information became available, we matched the neurons identified with this approach to EM-reconstructed neurons to yield split-GAL4 drivers encompassing 110 cell types that connect with the DANs and MBONs (*Figure 3*). Several of the cell types originally selected by LM were found to be not directly connected with MBONs or DANs. Nevertheless, these lines can be valuable for other types of investigations. For example, one such line, SS32237, was found to exhibit robust female-female aggression when activated (*Schretter et al., 2020*).

In the hemibrain EM connectome, there are about 400 interneuron cell types that have over 100 total synaptic inputs from MBONs and/or synaptic outputs to DANs. Our newly developed collection of split-GAL4 drivers covers 30 types of these 'major interneurons' of the MB (*Supplementary file 3*). While this constitutes a small fraction, it includes cell types with remarkable connectivity patterns. For instance, CRE011, present as a single neuron per hemisphere, integrates over 2000 inputs from nine types of MBONs. This is the highest number of synaptic inputs from MBONs among all interneurons (*Figure 3C*). CRE011 provides cholinergic input to reward DANs (*Figure 2E*) and neurons in the lateral accessory lobe, a premotor center housing dendrites of multiple descending neurons (*Kanzaki et al., 1994*; *Namiki et al., 2018*). Another notable example is SMP108, which receives inputs from multiple glutamatergic MBONs and makes the highest number of cholinergic connections to DANs involved in appetitive memory (*Figure 3C*). We recently reported on SMP108's role in second-order conditioning (*Yamada et al., 2023*) and its upstream neurons labeled in SS33917 in transforming appetitive memory into wind-directed locomotion (*Aso et al., 2023*). *Supplementary file 3* contains connectivity

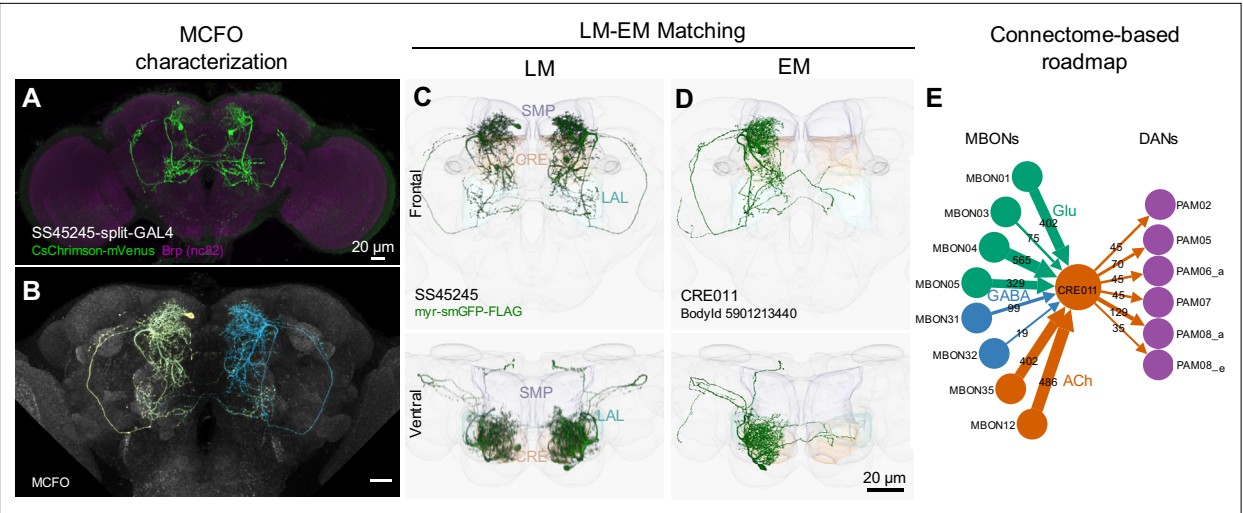

**Figure 2.** LM-EM Match of the CRE011-specific driver SS45245. (**A**) Expression pattern of SS45245-split-GAL4 in the brain. (**B**) MCFO image of SS45245 showing individual labeled neurons. (**C**) Frontal (top) and ventral (bottom) views of segmented light microscopy (LM) images of an exemplary split-Gal4 line (SS45245) visualized with a membrane reporter (myr-smGFP-FLAG) that was aligned to the JRC2018 standard brain. Projections are shown with outline of relevant neuropils. (**D**) The skeleton reconstructed from electron microscopy (EM) data of the matched cell type CRE011 in the hemibrain connectome. The CRE011 cell on the right hemisphere is shown. (**E**) Synaptic connectivity of CRE011 with MBONs and DANs in the MB derived from the hemibrain connectome.

The online version of this article includes the following figure supplement(s) for figure 2:

**Figure supplement 1.** LM-EM match of SS00460.

**Figure supplement 2.** LM-EM match of SS34963.

**Figure supplement 3.** LM-EM match of SS33915.

**Figure supplement 4.** LM-EM match of SS35020.

**Figure supplement 5.** LM-EM match of SS34979.

**Figure supplement 6.** LM-EM match of SS34947.

**Figure supplement 7.** LM-EM match of SS00486.

**Figure supplement 8.** LM-EM match of SS49308.

**Figure supplement 9.** LM-EM match of SS49361.

**Figure supplement 10.** LM-EM match of SS49868.

**Figure supplement 11.** LM-EM match of SS49352.

**Figure supplement 12.** LM-EM match of SS32228.

**Figure supplement 13.** LM-EM match of SS32219.

**Figure supplement 14.** LM-EM match of SS48890.

**Figure supplement 15.** LM-EM match of SS48799.

**Figure supplement 16.** LM-EM match of SS32259.

**Figure supplement 17.** LM-EM match of SS48341.

**Figure supplement 18.** LM-EM match of SS35040.

**Figure supplement 19.** LM-EM match of SS33905.

**Figure supplement 20.** LM-EM match of SS39538.

information of MBON-downstream and DAN-upstream neurons, along with the predicted neurotransmitters (*Eckstein et al., 2023*) and the available driver lines.

The current collection also contains over 180 lines for cell types that have innervations within the MB (*Figure 3—figure supplement 1*, *Supplementary file 4*, *Supplementary file 5*). These lines offer valuable tools to study several prominent cell types that previously are not genetically accessible. Notably, SS85572 enables the functional study of LHMB1, which forms a rare direct pathway from the calyx and the lateral horn (LH) to the MB lobes (*Bates et al., 2020*). SS48794 labels OA-VUMa2

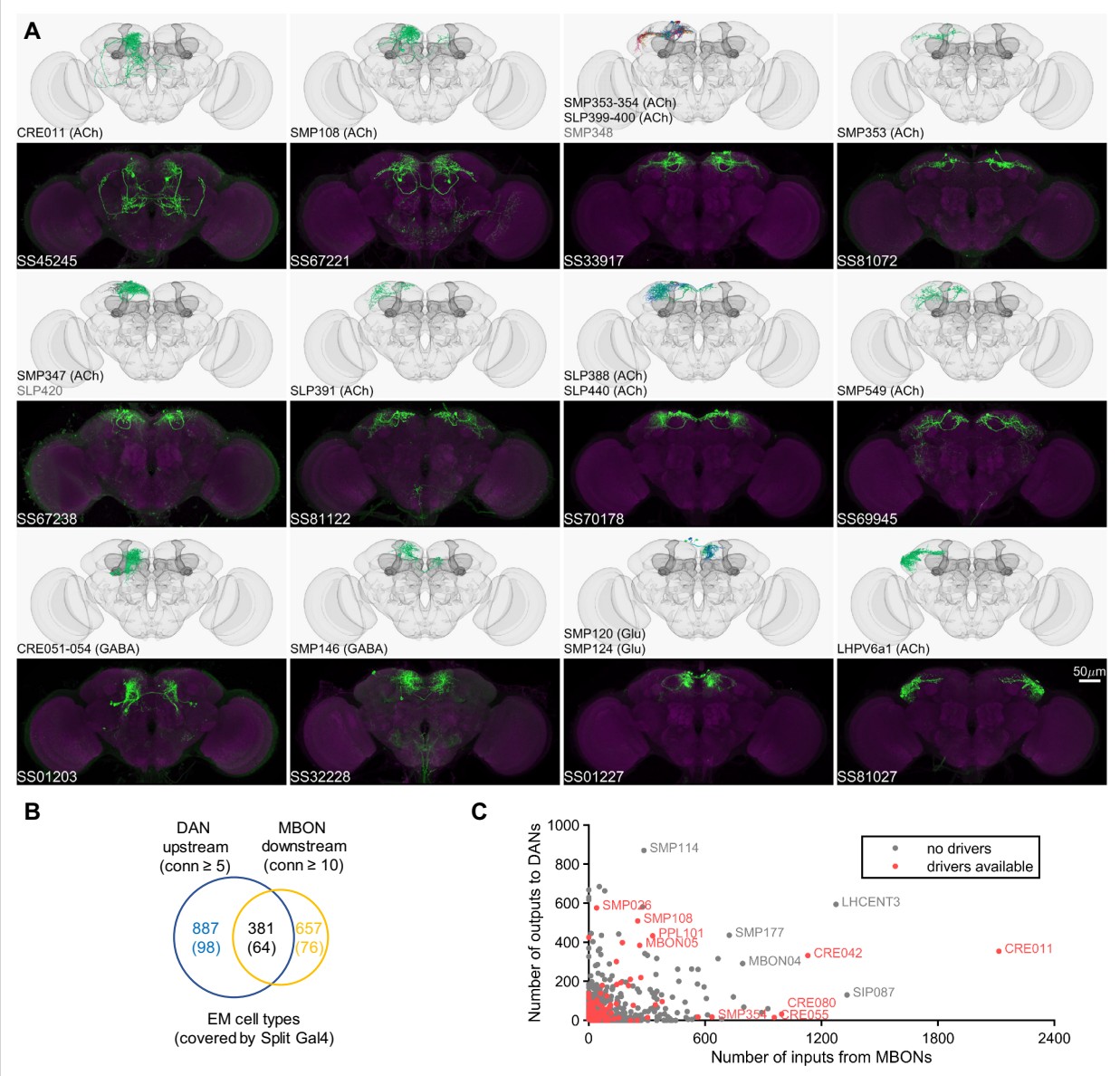

**Figure 3.** Drivers for MBON downstream and DAN upstream neurons. (**A**) Examples of confocal microscopy images of split-GAL4 lines (bottom) and their matching cell types in the hemibrain connectome (top). CsChrimson-mVenus (green); Brp (magenta). (**B**) The number of cell types that receive synaptic output from MBONs and supply synaptic input to DANs. Only cell types with connection (conn) over the indicated thresholds (i.e. more than 4 synapses for DAN upstream and more than 9 synapses for MBON downstream) were considered. The number of covered cell types are indicated in the brackets. (**C**) A scatter plot of MB interneuron cell types connected with DANs and MBONs. Cell types covered by Split-GAL4 lines are highlighted in red.

The online version of this article includes the following figure supplement(s) for figure 3:

**Figure supplement 1.** New or improved drivers for MB cell types.

octopaminergic neurons, which are the *Drosophila* counterparts to the honeybee OA-VUMmx1 neurons, the first neurons identified as mediating US signals in an insect brain (*Hammer, 1993*). Moreover, several drivers in this collection provide improved specificity. When combined with previous collections (*Aso et al., 2014a*; *Rubin and Aso, 2024*), we now have coverage for seven types of Kenyon cells and 62 out of 87 total cell types within the MB (excluding PNs). Overall, this amounts to over 70% coverage for non-PN cell types within the MB and about 10% coverage for MBON-downstream and DAN-upstream cell types (*Supplementary file 3*, *Supplementary file 5*, *Figure 3B and C*).

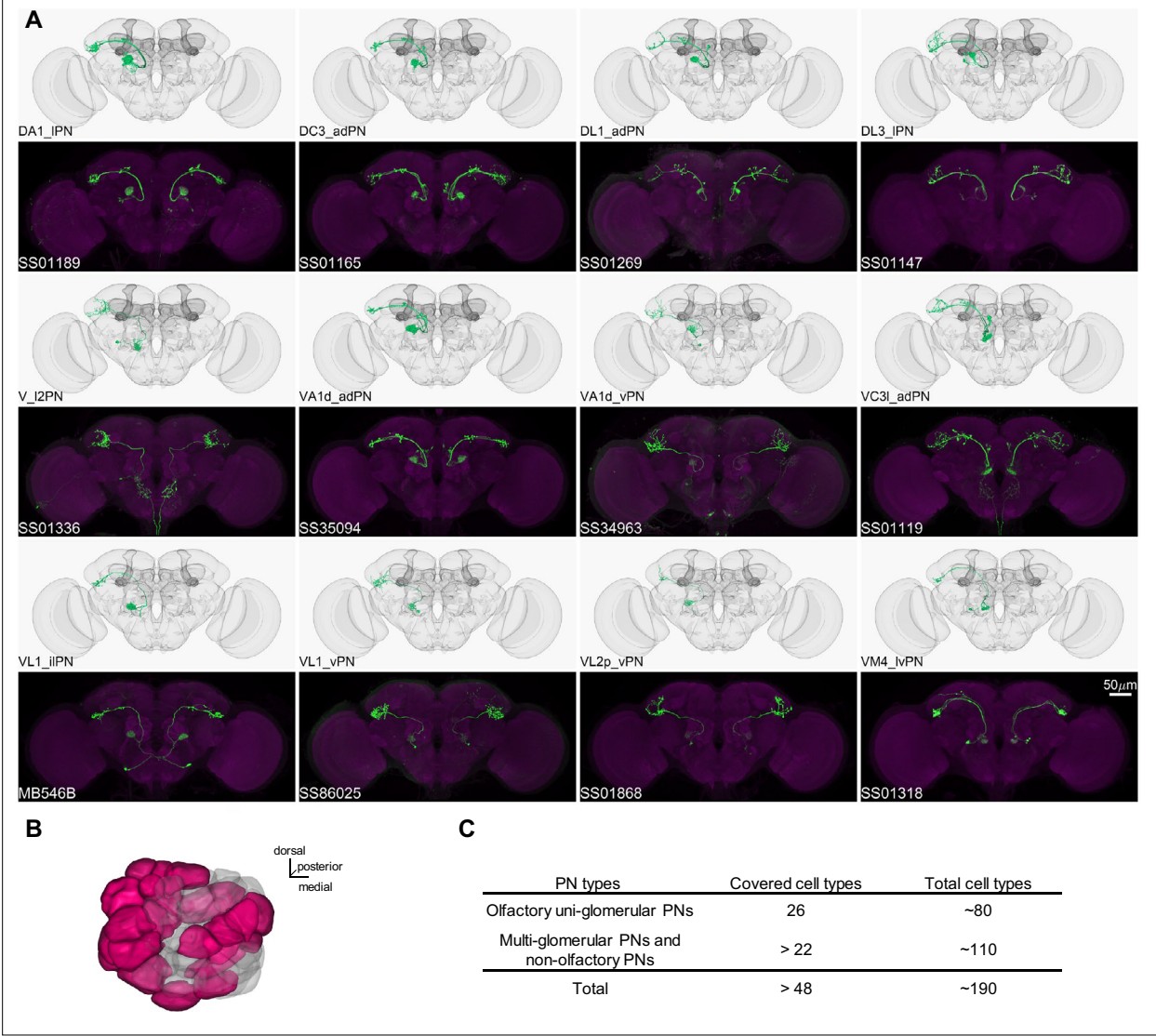

**Figure 4.** Driver lines for uni-glomerular projection neurons. (**A**) Examples of covered uni-glomerular PN (uPN) cell types. (**B**) Coverage of the 51 antennal lobe glomeruli. The new collection of split-GAL4 covers uPN in the colored glomeruli. (**C**) Split-GAL4 coverage summary.

| PN types | Covered cell types | Total cell types |
|---|---|---|
| Olfactory uni-glomerular PNs | 26 | ~80 |
| Multi-glomerular PNs and non-olfactory PNs | > 22 | ~110 |
| Total | > 48 | ~190 |

## Drivers for the antennal lobe projection neurons

In *Drosophila*, the primary CS pathway to the MB involves the antennal lobe PNs that convey olfactory signals. We have developed a set of driver lines for PNs and other cell types in the antennal lobe (*Supplementary file 1*). This set includes 191 lines, covering more than 48 of the approximately 190 PN types identified through EM connectome and LM studies (*Bates et al., 2020*; *Li et al., 2020*; *Lin et al., 2007*; *Tanaka et al., 2004*; *Zheng et al., 2022*). This set encompasses both uni- and multiglomerular PNs (*Figures 4 and 5*; *Supplementary file 6*).

The antennal lobe, in addition to the 51 olfactory glomeruli, contains 7 glomeruli involved in processing thermo- and hygro-sensory information (*Enjin et al., 2016*; *Frank et al., 2015*; *Gallio et al., 2011*; *Jenett et al., 2012*; *Liu et al., 2015*; *Marin et al., 2020*; *Stocker et al., 1990*; *Tanaka et al., 2012*). We provide 8 lines that cover sensory neurons projecting into these non-olfactory glomeruli and 18 lines covering the projection neurons emanating from them (*Figure 6*; *Supplementary file 1*, *Supplementary file 6*).

Although less abundant than the olfactory input, the MB also receives visual information from the visual projection neurons (VPNs) that originate in the medulla and lobula and project to the accessory

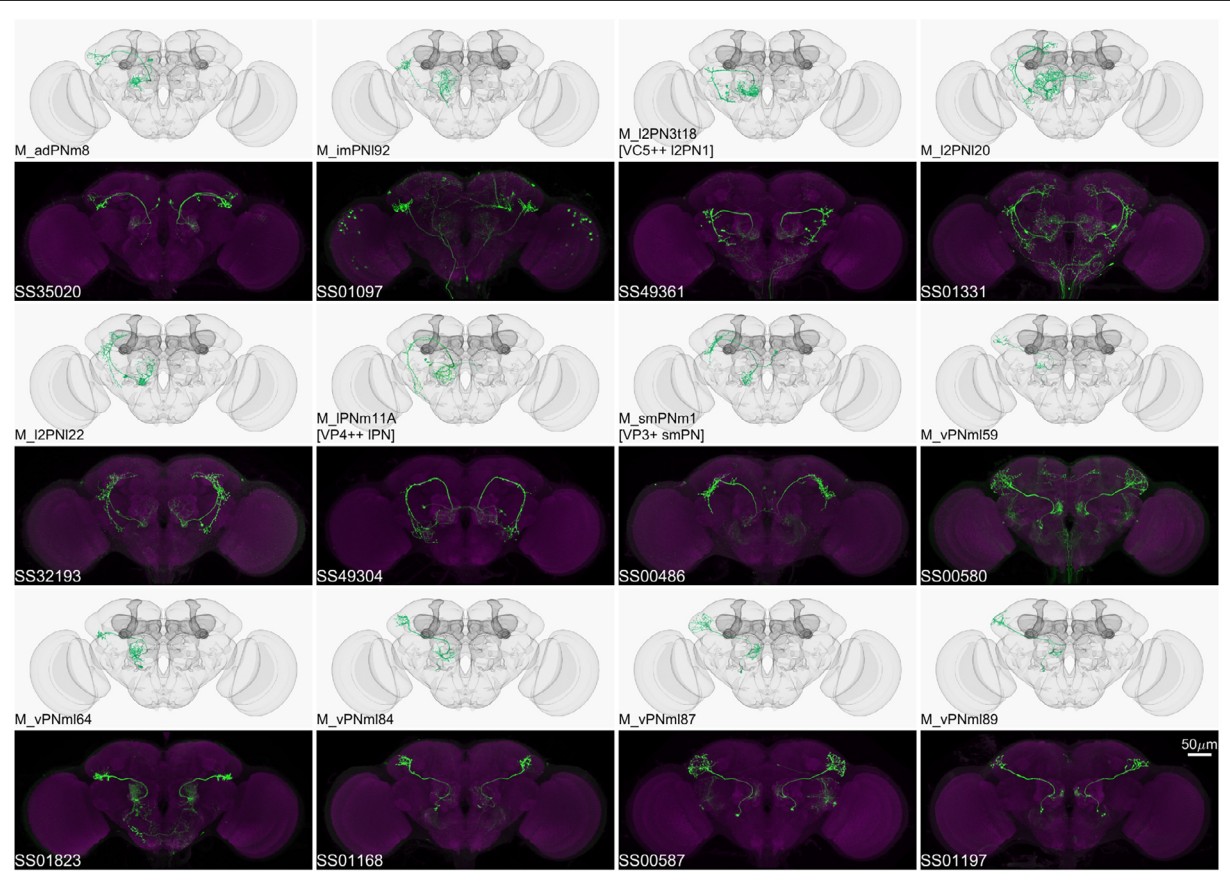

**Figure 5.** Driver lines for multi-glomerular projection neurons. Examples of multi-glomerular projection neurons (mPN) types. M_l2PN3t18 [VC5++ l2PN1], M_lPNm11A [VP4++ lPN], and M_smPNm1 [VP3+ smPN] are predicted to receive majority non-olfactory input (**Marin et al., 2020**).

calyx (**Li et al., 2020**; **Vogt et al., 2016**). A recent preprint described the full collection of split-GAL4 driver lines in the optic lobe, which includes the VPNs to the MB (**Nern et al., 2024**).

## Drivers for reinforcement pathways

Our understanding of the neural pathways that encode the US has been greatly advanced by experiments that have tested the sufficiency of various neuronal cell types to substitute for the US (**Aso et al., 2010**; **Aso and Rubin, 2016**; **Burke et al., 2012**; **Chiang et al., 2011**; **Claridge-Chang et al., 2009**; **Hige et al., 2015a**; **Huetteroth et al., 2015**; **Liu et al., 2012**; **Saumweber et al., 2018**; **Schroll et al., 2006**; **Yamagata et al., 2015**). These experiments leveraged thermogenetic or optogenetic tools expressed in specific neuronal cell types, especially DANs, to assess their functions in associative learning. The approach to directly stimulate DANs, although valuable, bypasses the earlier US processing pathways and potential feedback in the circuit. Because of this experimental caveat, it is preferable to activate neurons at the sensory level of reward or punishment pathways to faithfully replicate the natural activity of these DANs. In that way, DANs can be regulated by both US sensory pathways and feedback pathways from MBONs. That is likely to be essential for the successful execution of more complex learning tasks in which flies update memories based on the current and past experiences (**Felsenberg et al., 2018**; **Felsenberg et al., 2017**; **Jiang and Litwin-Kumar, 2021**; **McCurdy et al., 2021**; **Otto et al., 2020**; **Rajagopalan et al., 2022**).

Our collection identified several useful genetic tools that advance this approach further. For example, we generated drivers for the cell types in the thermo-sensory pathways, as well as PNs for odors with innate preference, such as $CO_2$ and cVA (**Datta et al., 2008**; **Lin et al., 2013**; **Suh et al., 2004**). These cell types are candidates to convey the reinforcement signals to the MB and other brain regions for associative learning.

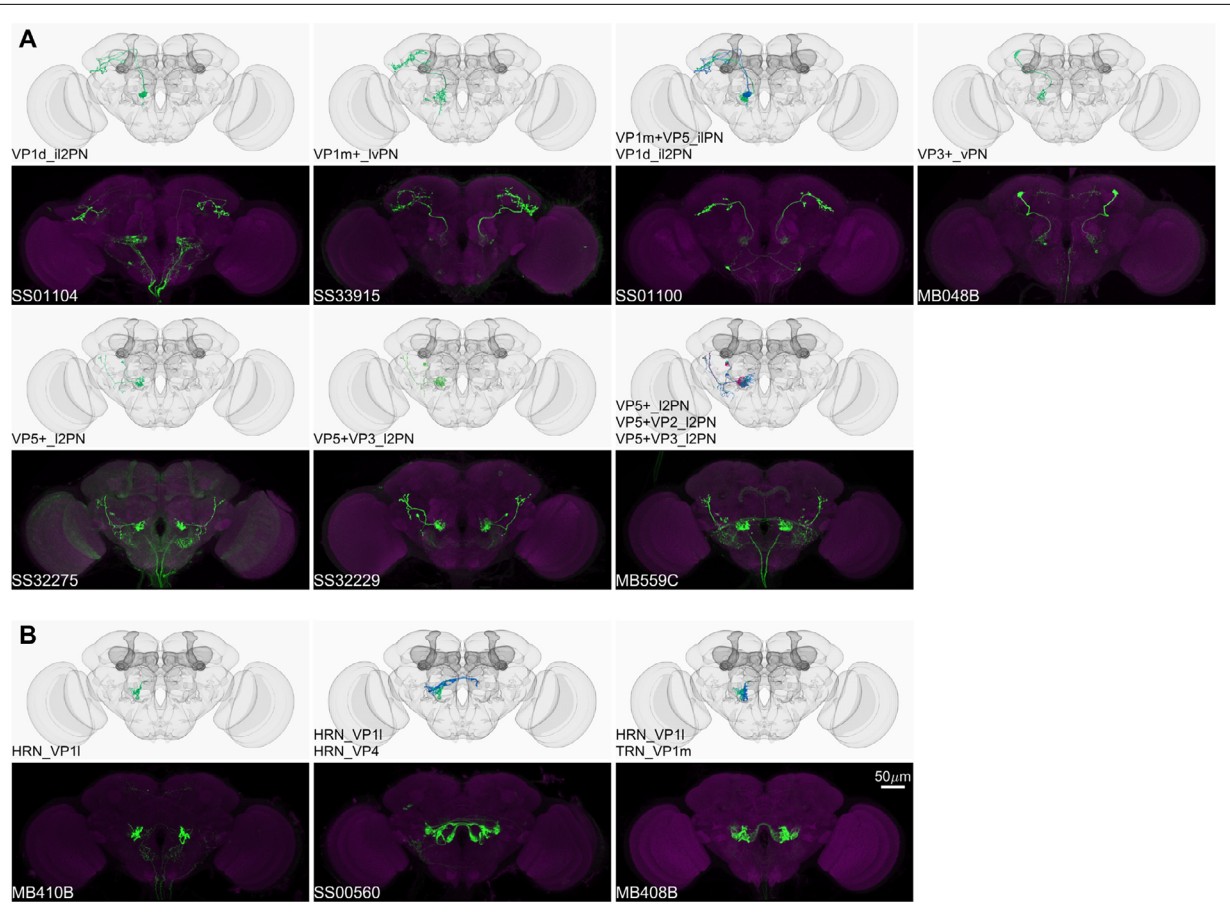

**Figure 6.** Driver lines for non-olfactory projection neurons and sensory neurons. Examples of thermo-/hygro-sensory PN types (**A**) and sensory neurons (**B**) covered by this collection. Other than thermo-/hygro-sensory receptor neurons (TRNs and HRNs), SS00560 and MB408B also label olfactory receptor neurons (ORNs): ORN_VL2p and ORN_VC5 for SS00560, ORN_VL1 and ORN_VC5 for MB408B.

We had a particular interest in developing a driver panel for gustatory sensory neurons. Although they play a central role in reward signaling, they convey those signals to the MB through largley uncharacterized pathways (*Bohra et al., 2018*; *Burke et al., 2012*; *Deere et al., 2023*; *Kim et al., 2017*; *Miyazaki et al., 2015*; *Sterne et al., 2021*). GAL4 driver lines that recapitulate expression patterns of gustatory receptors (GRs) have been generated and utilized for functional studies (*Dahanukar et al., 2007*; *Harris et al., 2015*; *Miyamoto et al., 2012*; *Wang et al., 2004*; *Yavuz et al., 2014*). However, these driver lines tend to contain a morphologically and functionally heterogeneous set of sensory neurons (see for examples: *Chen et al., 2022*; *Thoma et al., 2016*) and may have off-target expression. To address these limitations, we have developed split-GAL4 drivers specific to different subsets of gustatory sensory neurons by generating hemidrivers for GR-gene promoters and screening intersections with existing hemidrivers (*Figure 7A*).

In fruit flies, sugar is detected by sensory neurons located on different taste organs of the body and also inside the brain (*Fujii et al., 2015*; *Hiroi et al., 2002*; *Miyamoto et al., 2012*; *Rodrigues and Siddiqi, 1978*). Gr64f-Gal4, in which Gal4 is expressed under the control of the promoter of the sugar receptor gene *Gr64f*, broadly labels sugar sensory neurons (*Dahanukar et al., 2007* and *Figure 7—figure supplement 1*). Gr64f-Gal4 expression can be found in heterogeneous sets of sensory neurons in the labellum, the tarsi and the labral sense organ (LSO) located along the pharynx. In addition, Gr64f-Gal4 also labels subsets of olfactory receptor neurons and neurons innervating the abdominal ganglion (*Park and Kwon, 2011*; *Figure 7—figure supplement 1*). Whether these cells endogenously express Gr64f is yet to be confirmed. However it seems likely that the heterogeneity of Gr64f-Gal4 expression could limit its usage in generating fictive rewards in complex associative learning.

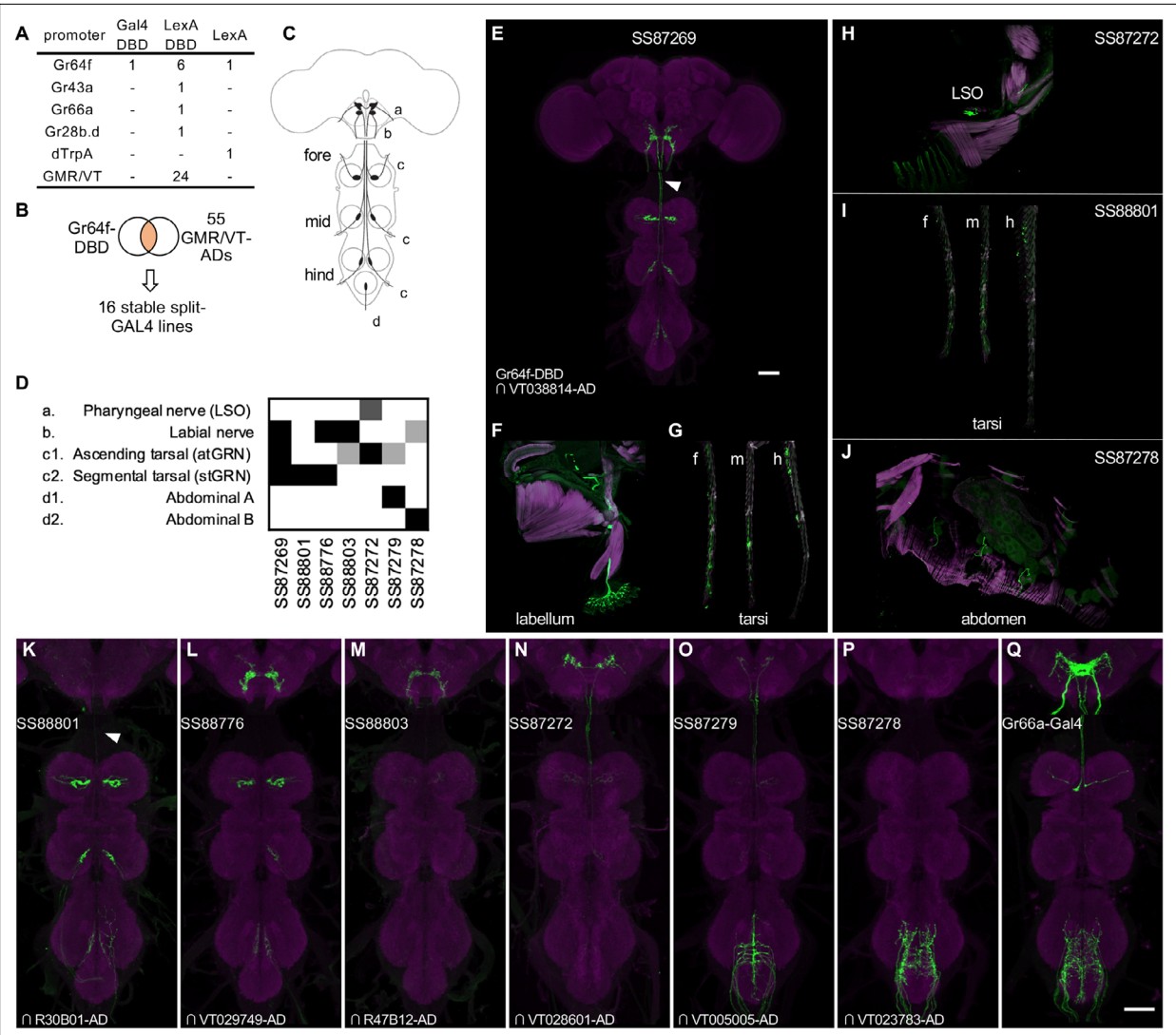

**Figure 7.** Driver lines for gustatory sensory neurons. (**A**) A summary of new transgenic lines generated by this study. In addition to Gr64f promoters, LexADBD lines were generated with Gr64f, Gr43a, Gr66a, Gr28b promoters and other GMR or VT promoters. The six Gr64f-LexADBD lines are with different insertion sites, and with the presence or absence of the p10 translational enhancer (see **Supplementary file 2** for details). (**B**) Schematic of the screening strategy used here to subdivide the Gr64f-DBD pattern by intersecting it with GMR and VT AD lines. (**C**) A schematic of the sensory neuron projection types that compose the Gr64f-DBD pattern. (**D**) A summary of expression patterns of 6 of the Gr64f split-GAL4 lines derived from the screening strategy in (**B**), using the anatomical notation described in (**C**). (**E**) Expression pattern of SS87269 in the brain and VNC. The arrow indicates an ascending projection of atGRN. (**F**) Expression of SS87269 in the labellum. (**G**) Expression of SS87269 in tarsi of fore (f), middle (m) and hind (h) legs. (**H**) Expression of SS87272 in the labral sense organ (LSO). (**I**) Expression of SS88801 in tarsi. (**J**) Expression of SS87278 in the abdominal body wall. (**K–Q**) Expression patterns of designated driver lines in the Gnathal Ganglia (GNG) and VNC. The arrow in K indicates the absence of ascending projections from stGRN. Magenta in F-J indicates muscle counterstaining with phalloidin (actin filaments); magenta in other panels indicates neuropil counterstaining of Brp. All scale bars are 50 μm.

The online version of this article includes the following figure supplement(s) for figure 7:

**Figure supplement 1.** Expression pattern of Gr64f-Gal4.

**Figure supplement 2.** Expression pattern of Split-LexA lines from Gr43a and Gr66a.

**Figure supplement 3.** Examples of covered SEZ neurons.

To refine Gr64f-Gal4 expression, we intersected Gr64f-GAL4DBD with various AD lines selected to have overlapping expression with a subset of the projection patterns of the original Gr64f-GAL4. We obtained 16 stable split-GAL4 lines with labeling in distinct subsets of the original expression pattern (**Figure 7**). We examined the ability of these lines to serve as US in olfactory learning (**Aso and Rubin,**

*2016*) and their potency to drive local search behaviors, another memory-based behavior induced by appetitive stimuli (*Corfas et al., 2019*; *Figure 8*, *Figure 8—figure supplement 1*). Additionally, we measured the walking speed of flies, as flies decrease walking while feeding (*Thoma et al., 2016*; *Aso and Rubin, 2016*).

Among the Gr64f-split-GAL4 lines, SS87269 emerged as the best driver for substituting sugar reward in olfactory learning given its anatomical specificity as well as its effectiveness throughout long training protocols (*Figure 8E*). SS87269 expresses in the labellum and in at least two types of tarsal sensory neurons, namely the ascending (atGRN) and the non-ascending segment-specific (stGRN) types (*Figure 7E and F*). The driver does not label pharyngeal sensory neurons, and importantly, it lacks expression in abdominal ganglion and olfactory sensory neurons, which could reflect off-target expression from the original Gr64f-GAL4. When odor presentation was paired with the activation of SS87269 neurons, flies formed robust appetitive memories even after extended training with high LED intensity (*Figure 8E*, *Figure 8—figure supplement 1* and *Video 1*). Furthermore, and consistent with the idea that this subset of sensory neurons encodes appetitive food-related taste information, activating these neurons elicited proboscis extension and slowed flies (*Figure 8F–I*; *Figure 8—figure supplement 2*, *Video 2* and *Video 3*). These flies also showed robust local search behavior during the post-activation period, that is an increased probability of revisiting the area where they experienced the activation (*Video 4*). Notably, the revisiting phenotype of SS87269 was stronger than any other Gr64f-split-GAL4 drivers (*Figure 8*, *Figure 8—figure supplement 2Figure 8F*, *Figure 8—figure supplement 2A*) and the original Gr64f-GAL4 even after 180 repetitive activation trials (*Figure 8—figure supplement 3*).

Two other lines SS88801 and SS88776, which label stGRNs or stGRNs along with labial sensory neurons, respectively (*Figure 7K–L and I*), similarly showed appetitive learning and reduced locomotion during activation (*Figure 8E–F*). Interestingly, however, the activation of stGRNs with SS88801 did not induce significant local search behaviors (*Figure 8F*, *Figure 8—figure supplement 2A*). This finding could be valuable for understanding circuits underlying local search behavior and invites further investigation to compare pathways from labial and tarsal sensory neurons to the MB and the central complex.

In contrast to SS87269, two other lines, SS87278 and SS87279, express in cells that appear to convey aversive signals. Activation of these lines induced an increase in walking speed during activation and reduced the probability of revisiting where flies received activation LED light (*Figure 8F*, *Figure 8—figure supplement 2*). Also, flies became progressively less mobile during the inter-trial interval period (*Figure 8G*). We made a similar observation with the original Gr64f-GAL4 (*Figure 8G*) as well as with a bitter-taste driver, Gr66a-GAL4 (data not shown). With extended training using SS87278 and SS87279, the preference to the paired odor eventually became negative (*Figure 8E*). These drivers label distinct subsets of sensory neurons projecting to the abdominal ganglion (*Figure 7O and P*). The innervation of SS87278 inside the abdominal ganglion is similar to that of Gr66a-GAL4 (*Figure 7Q*; *Dunipace et al., 2001*), which is known to label multidendritic sensory neurons in the adult *Drosophila* abdomen (*Shimono et al., 2009*). Examining projection patterns in fly bodies with whole-animal agar sections, we found that sensory neurons in SS87278 also project to the abdominal body surface (*Figure 7J*), likely representing the class IV multidendritic neurons that detect multimodal nociceptive stimuli (*Hwang et al., 2007*; *Ohyama et al., 2015*). The expression in these aversive sensory neurons therefore may complicate the interpretation of appetitive learning with the original Gr64f-GAL4.

Overall, the refinement of Gr64f-Gal4 expression with SS87269 now allows for specific manipulation of the rewarding subset of gustatory sensory neurons and permits training with an extended number of trials. While we have not yet conducted extensive anatomical screening, LexADBD lines generated with Gr64f, Gr43a, Gr66a, Gr28b.d promoters (*Figure 7A*) should allow for comparable refinement of driver lines to target sensory neurons expressing these receptors. For instance, we generated split-LexA drivers specific for internal fructose sensory neurons (*Miyamoto et al., 2012*) or a subset of bitter taste neurons (*Figure 7—figure supplement 2*). We also made a small number of lines for cell types in the subesophageal zone (SEZ) area (*Figure 7—figure supplement 3*), which complement previous collections of drivers for gustatory interneurons (*Otto et al., 2020*; *Sterne et al., 2021*).

Lastly, we generated driver lines for putative ascending nociceptive pathways. We tested whether optogenetic activation would drive avoidance related behaviors using the circular arena, where an

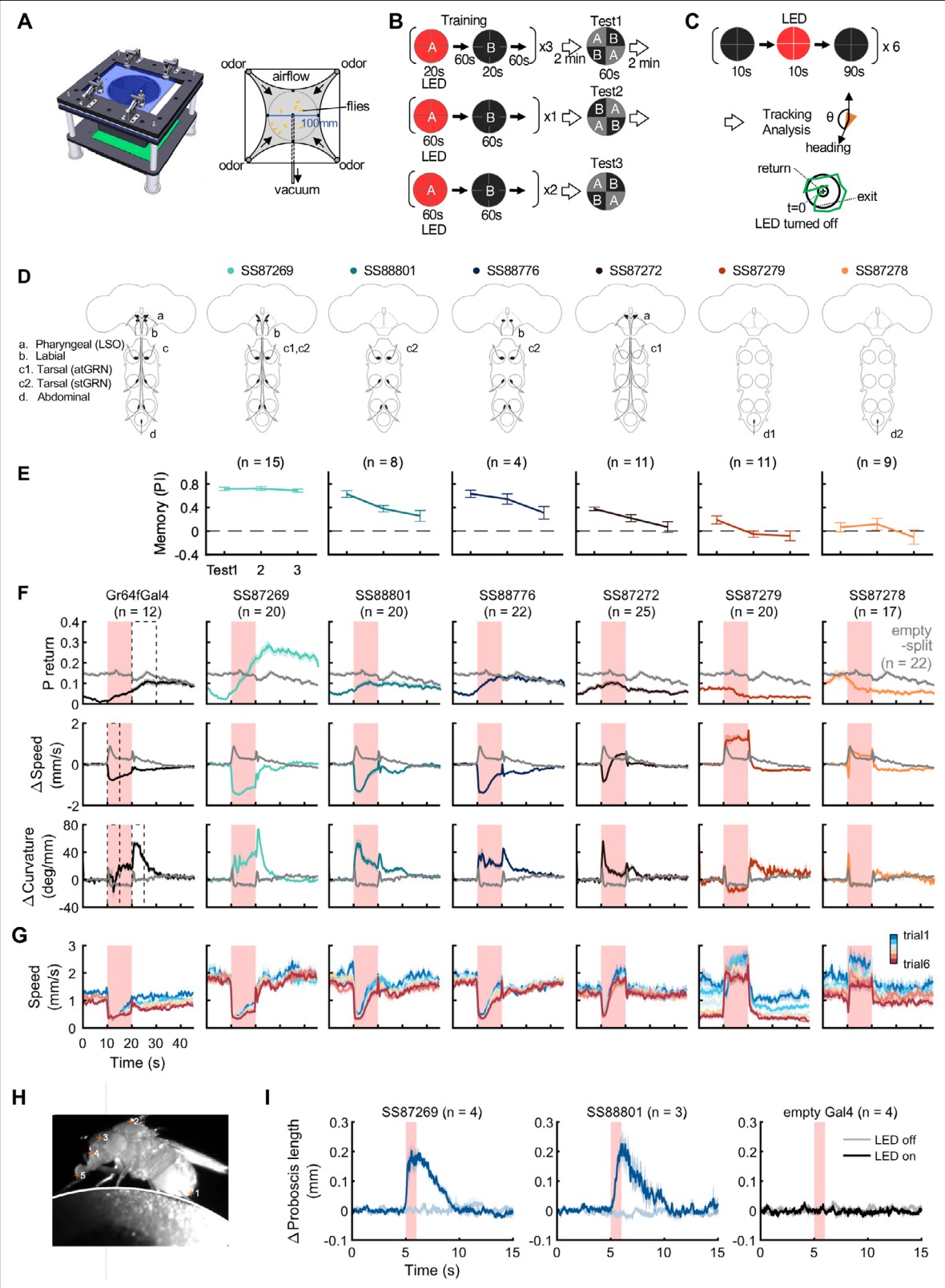

**Figure 8.** Behaviors with Gr64f-split-GAL4 lines. (**A**) In the optogenetic olfactory arena, odors are delivered from the periphery and sucked out from the central vacuum port to deliver odors only to defined quadrants. Red LEDs (627 nm peak) enable programmed optogenetic activation of neurons expressing CsChrimson. (**B**) Training and testing protocol for experiments shown in (**E**). The training protocol consisted of 3x20s optogenetic activation training followed by the first preference test, 1x1 min training followed by the 2nd test, and additional 2x1 min training followed by the last test. Odors

*Figure 8 continued on next page*

*Figure 8 continued*

delivered to the two zones and odor durations in each period are indicated. LED intensities were 4.3 µW/mm² in early 20 s training and 34.9 µW/mm² in later training. Activation LED was applied with 1 s ON and 1 s OFF during pairing with odor A. Odors were trained reciprocally. Pentyl acetate and Ethyl lactate were used as odor A and B, respectively, in one half of the experiments and the two odors were swapped in the other half of experiments. (**C**) Protocol to characterize Gr64f split-GAL4 activation phenotypes in the absence of an odor. During each trial, flies were illuminated with a red LED light continuously for 10 s. (**D**) Summary diagram of the expression patterns of the original Gr64f-GAL4 (far left) and 6 Gr64f-split-GAL4s. The expression of the original Gr64f-GAL4 in olfactory sensory neurons is not depicted here. (**E**) Associative memory scores after the training protocol in (**B**). Mean, standard error of the mean (SEM), and the number of groups are shown. (**F**) The kinematic parameters of trajectories measured with Caltech FlyTracker during split-GAL4 activation in the absence of odor as shown in (**C**). Return behavior was assessed within a 15 s time window. The probability of return (P return) is the number of flies that made an excursion (>10 mm) and then returned to within 3 mm of their initial position divided by the total number of flies. Curvature is the ratio of angular velocity to walking speed. Each group of flies received 6 activation trials. Summarization was based on the trial average of each group. The number of groups is indicated. The thick lines and shadows are mean and SEM. Gray lines are Empty-split-GAL4 control. Dashed lines are time bins for data summary in *Figure 8—figure supplement 2*. (**G**) Average walking speed in each of 6 trials. (**H**) An image of a tethered fly on a floating ball. Flies were tracked for proboscis extension (PE) activity with the Animal Part Tracker (*Kabra et al., 2022*). The annotated points, in the order of numbers, consisted of the tip of the abdomen (1), the highest point on the thorax (2), the midpoint between the root of the antennae (3), the base of the proboscis (4) and the tip of the proboscis (5). PE activity was quantified as the change of proboscis length, i.e., the distance from the tip to the base of the proboscis, or the distance between points 4 and 5. (**I**) SS87269 and SS88801 activation and proboscis extension. Each fly was recorded over 6 activation trials in which the 624 nm LED was turned on for 1 s. LED intensity for SS87269 and SS88801, 11 µW/mm²; for empty Gal4 (pBDPGal4), 50 µW/mm². Less saturated traces indicate behavior during LED off trials, while more saturated traces indicate behavior during LED on trials.

The online version of this article includes the following source data and figure supplement(s) for figure 8:

**Source data 1.** The numerical values to generate the plots in *Figure 8E–G and I*.

**Figure supplement 1.** Olfactory arena learning experiment, fly tracking and data analysis.

**Figure supplement 1—source data 1.** The numerical values to generate the plots in *Figure 8—figure supplement 1*.

**Figure supplement 2.** Summary data of Gr64f-split-Gal4 activation phenotypes.

**Figure supplement 2—source data 1.** The numerical values to generate the plots in *Figure 8—figure supplement 2*.

**Figure supplement 3.** Consistency of Gr64f-split-GAL4 phenotypes over repeated activations.

activating LED illuminated two quadrants of the arena, and the other two remained dark. We determined the quadrant preference for 581 combinations of ZpGAL4DBD and p65ADZp hemidrivers. We found one driver, SS01159, which showed the most robust avoidance of the illuminated quadrants, and exhibited behavioral features characteristic of nociception, including backward

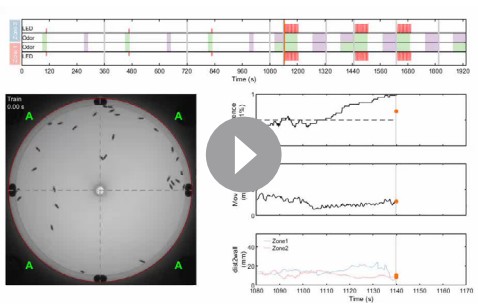

**Video 1.** Olfactory arena learning experiment with SS87269. An exemplar video of learning of flies of the genotype SS87269/UAS-CsChrimson-mVenus attP18. Movies were tracked with Caltech FlyTracker (*Eyjolfsdottir et al., 2014*). Trailing trajectories of individual flies in the last 5 s were overlaid. Delivery of odor A and B to the quadrants along with the 625 nm LED activation are indicated. Experiment movie and data from the first 1x1 m training and test are presented with the gray line indicating session transition.

https://elifesciences.org/articles/94168/figures#video1

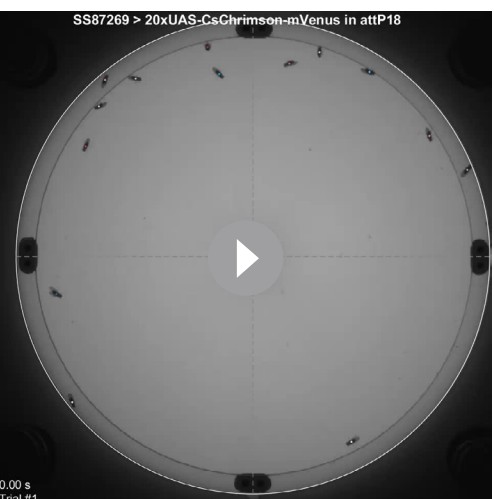

**Video 2.** Activation phenotypes with SS87269. An exemplar video of activation of flies of the genotype SS87269/UAS-CsChrimson-mVenus attP18. Flies receive six consecutive 60 s trials; during each trial a 10 s LED activation was presented (from 10 to 20 s) as indicated. The trajectories of individual flies over the previous 5 s are shown.

https://elifesciences.org/articles/94168/figures#video2

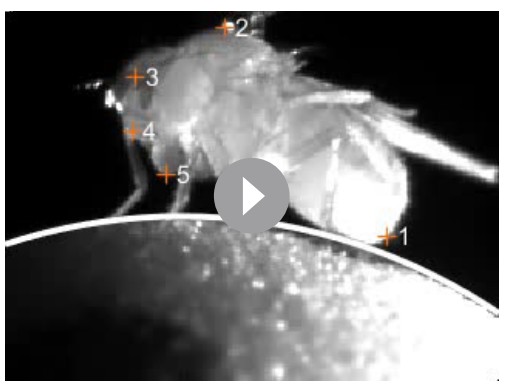

**Video 3.** Proboscis extension upon SS87269 activation for flies walking on a ball. Flies were thorax-fixed by tethering to a pin and allowed to walk on an air-floated foam ball. Proboscis activities were tracked with the Animal Part Tracker (https://github.com/kristinbranson/APT; *Branson, 2024*; *Kabra et al., 2022*).
https://elifesciences.org/articles/94168/figures#video3

walking, turning, increased speed, and jumping upon activation (*Figure 9B–D*, and *Video 5*). This driver labels a group of ascending neurons (*Figure 9E*), which likely carry nociceptive signals from the body and legs to the brain. We then generated drivers for subsets of these ascending neurons guided by single neuron morphologies of cells in SS01159 determined by MCFO stochastic labeling (*Nern et al., 2015*). The collection in total contains approximately 100 split-GAL4 lines covering ascending neurons. While not completely matched to EM cell types due to only a portion of their morphologies being available in the hemibrain volume, these lines serve as a valuable resource for querying the reinforcement pathways.

## Morphological individuality and asymmetry

Neuronal morphology can vary, and the randomness of developmental processes can ultimately result in differences in behavior among individuals (Linneweber). The recent landmark study systematically examined morphological variability of neurons using the EM connectomic data (*Schlegel et al., 2023*), but it will only be possible to examine relatively few individual brains for the foreseeable future. As a part of Janelia Flylight team project to generate cell-type-specific driver lines, we have imaged over 6,000 fly brains for the present study. While annotating those confocal images, we occasionally encountered samples with atypical morphologies (*Figure 10*). For example, one V_l2PN neuron, which typically projects to the lateral horn of the ipsilateral hemisphere, exhibited a peculiar morphology in one brain sample, where its axon crossed the midline and projected to the contralateral hemisphere, yet it still reached the correct target area within the lateral horn of the opposite side (*Figure 10A*). Another instance involved a DPM neuron, the sole serotonergic neuron of the MB lobes. While typical brain samples contain only one DPM neuron per hemisphere, we found a brain with two DPM neurons in one hemisphere (*Figure 10B*). In this case, the DPM projections exhibited an atypical innervation of the calyx of the mushroom body. We also found examples involving MBONs. The dendrites of MBON-α1 are typically confined to the α lobe of the MB, but we discovered a case where this cell in addition sent projections to the ellipsoid body (*Figure 10C*). MBON-α3 displayed striking variability in soma positions, but only subtle variability in axonal projections (*Figure 10D*). The table in *Figure 10E* summarizes additional examples of the atypical morphologies of MBONs. Overall in 1241 brain hemispheres examined, we found mislocalization of dendrites and axons in 3.14% and 0.97% of MB cell types, respectively. If this rate of mislocalization is generalizable to other brain regions, a fly brain of ~130,000 neurons (*Dorkenwald et al., 2023*) would have a few thousands of neurons with mislocalized dendrites or axons. These examples of atypical morphology were observed only once in dozens of samples, and thus can be considered as erroneous projections either resulting from stochastic developmental processes, or possibly caused by ectopic expression of reporter proteins on the plasma membrane at a high level.

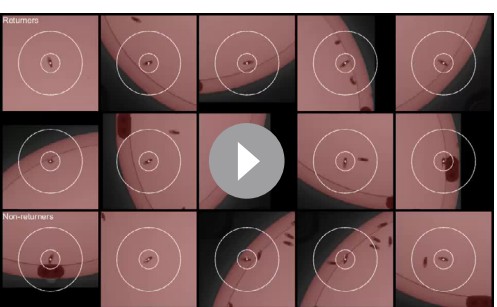

**Video 4.** Return phenotype of SS87269 at LED offset. Behaviors of individual flies with their trajectories from LED offset up to 15 s after LED offset. Videos are centered to the positions of flies at LED offset. The two white circles indicate 3 mm and 10 mm boundaries from the position. Flies were sorted by the time they re-entered the 3 mm inner circle after they exited the 10 mm outer circle. Flies at the bottom row did not return within the 15 s time frame.
https://elifesciences.org/articles/94168/figures#video4

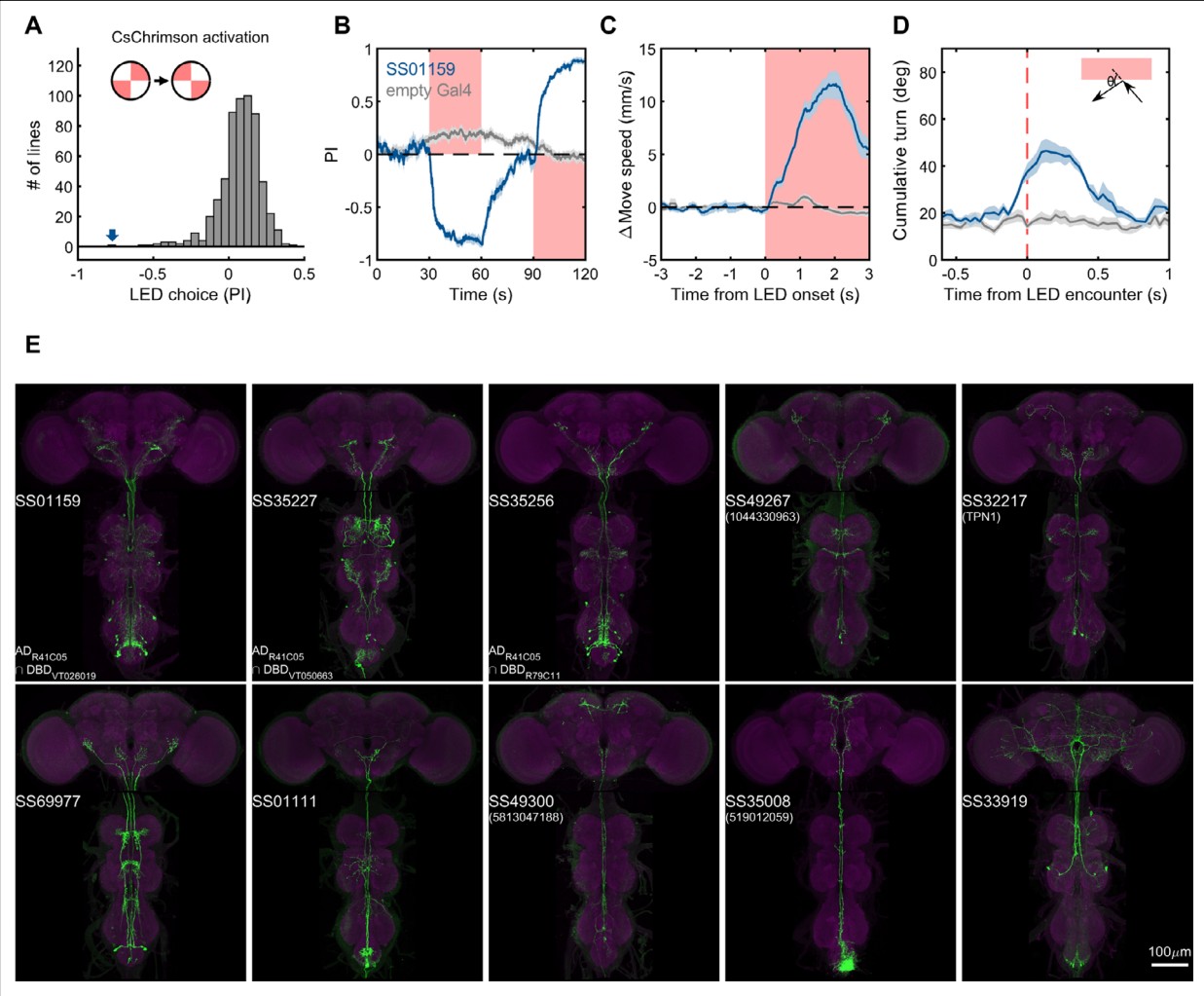

**Figure 9.** Examples of covered ascending neurons. (**A**) Activation preference screen of 581 split-GAL4 lines (342 lines from this study). SS01159 (blue arrow) is one of the lines that flies showed strong avoidance at optogenetic activation. (**B**) Time course of flies' preference to quadrants with red LED light by SS01159>CsChrimson (blue) or empty-GAL4>CsChrimson (gray). A preference score to red LED quadrants was quantified from the distribution during the last 5s of two 30s activation periods. n = 8 groups for SS01159, n = 15 for empty Gal4. Mean (thick lines) and SEM (shadow) are plotted. (**C**) The mean normalized movement speed at the LED onset for flies in the LED quadrants. The 3-s period before LED onset was used as the baseline for normalization. (**D**) The mean cumulative turning angles in 5 movie frames (total elapsed time of 167 ms) when flies encountered the LED boundary. The boundary was defined as a 4-mm zone in between the LED and dark quadrants. Trajectories too close to the center (< 0.2*radius) or the wall (> 0.6*radius) of the arena were not analyzed. (**E**) Examples of split-GAL4 lines for ascending neurons. SS35227 and SS35256 shared a split half (R41C05-AD) with SS01159. SS32217 matched with TPN1 (***Kim et al., 2017***). No cell types (or only EM BodyIds) were assigned to the other lines shown due to missing information in the hemibrain volume.

The online version of this article includes the following source data for figure 9:

**Source data 1.** The numerical values to generate the plots in ***Figure 9B–D***.

In contrast to these rare, and seemingly erroneous, morphological variations, we observed much more frequent and reproducible variations in the composition as well as morphologies in the two MBONs (MBON08 and MBON09) labeled by MB083C, which may amount to 'individuality'. This split-GAL4 driver line invariably labels two cells in each hemisphere in 169 brain samples examined with four different reporters and in both sexes (57 males and 112 females; ***Figure 11—figure supplement 1***). In all samples, these MBONs arborize dendrites in the γ3 and β'1 compartments. An obvious mistargeting of the axon was observed in only one sample, suggesting highly consistent development of these MBONs (***Figure 11A and B***). However, MCFO method visualized two distinct morphologies of these MBONs: MBON08 arborizes dendrites in the γ3 compartment of both hemispheres, whereas MBON09 arborize dendrites in ipsilateral γ3 and contralateral β'1 (***Figure 11C–H***; ***Aso et al.,***

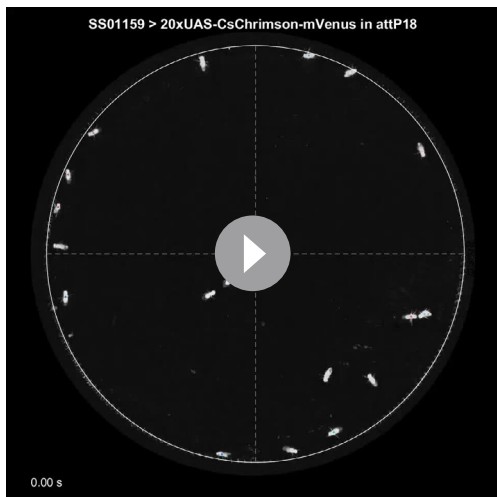

**Video 5.** Activation preference of SS01159 in the LED choice experiment. An exemplar video of LED choices of flies of the genotype SS01159/UAS-CsChrimson-mVenus attP18. The trajectories of individual flies over the previous 5 s are shown. 625 nm LED was on from 30 s to 60 s and from 90 s to 120 s, and the lit quadrants are indicated.

https://elifesciences.org/articles/94168/figures#video5

*2014a*). β'1 compartment was always labeled in both hemispheres for all 169 samples, suggesting that MBON09 represents at least one of the two cells in each hemisphere. The second cell can be either MBON08 or MBON09. In MCFO experiments, we observed 21 instances of MBON08 (8 in the left and 13 in the right hemisphere) and 188 instances of MBON09 (*Figure 11I*). Based on these observations, we expect 65% of flies contain four MBON09, while the remaining 35% of flies likely have at least one MBON08 (*Figure 11J*). In 71 hemispheres, two cells were visualized in different colors of MCFO: 52 contained two MBON09 and 19 contained one MBON08 and MBON09. We never observed a brain with MBON08 on both hemispheres or two MBON08 in one hemisphere (*Figure 11J*). When MBON08 and MBON09 coexist, MBON09 arborized in the lateral part of the ipsilateral γ3 and MBON08 arborize in the medial part of the contralateral γ3 (*Figure 11E–H*). This seemingly extended γ3 compartment innervated by MBON08 is not part of γ4, because it did not overlap with DANs in the γ4 (*Figure 11—figure supplement 2A and B*).

Although MBON08 was not found in brain samples used for the hemibrain or FAFB EM connectome (*Dorkenwald et al., 2023*; *Scheffer et al., 2020*; *Zheng et al., 2018*), DANs in the γ3 could be subdivided to two groups that innervate the medial or lateral part of the γ3 (*Figure 11—figure supplement 2C*; *Li et al., 2020*). Therefore, subdivision of the γ3 compartment may exist irrespective of heterogeneity on the MBON side. In larval brains, two MBONs that correspond to adult MBON08/09 exhibit identical morphology (*Eichler et al., 2017*; *Saumweber et al., 2018*; *Truman et al., 2023*). During metamorphosis, these larval MBONs may acquire distinct morphology as MBON08 at 21/209 odds. We have never observed a brain with MBON08 on both hemispheres, and therefore MBON08 is likely to appear in only one hemisphere, if at all (*Figure 11J*). This resulting asymmetry could be one source of turning handedness and idiosyncratic learning performance (*de Bivort et al., 2022*; *Smith et al., 2022*), given that MBON09 forms extensive connections with other MBONs and fan-shaped body neurons (*Hulse et al., 2021*; *Li et al., 2020*) and the activity of MBON08/MBON09 has a strong influence on turning (*Aso et al., 2023*; *Matheson et al., 2022*).

## Conversion to split-LexA

Split-GAL4 lines enable cell-type-specific manipulation, but some experiments require independent manipulation of two cell types. Split-GAL4 lines can be converted into split-LexA lines by replacing the GAL4 DNA binding domain with that of LexA (*Ting et al., 2011*). To broaden the utility of the split-GAL4 lines that have been frequently used since the publication in 2014 (*Aso et al., 2014a*), we have generated over 20 LexADBD lines to test the conversions of split-GAL4 to split-LexA. The majority (22 out of 34) of the resulting split-LexA lines exhibited very similar expression patterns to their corresponding original split-GAL4 lines (*Figure 12*). The main mode of failure when converting to LexA was that expression levels become too weak and stochastic.

## Concluding remarks

The ability to define and manipulate a small group of neurons is crucial for studying neural circuits. Here, we have generated and characterized driver lines targeting specific cell types that are likely to be a part of associative learning circuits centered on the MB. We have provided these driver lines with a comprehensive lookup table linking these cell types with the EM hemibrain connectome

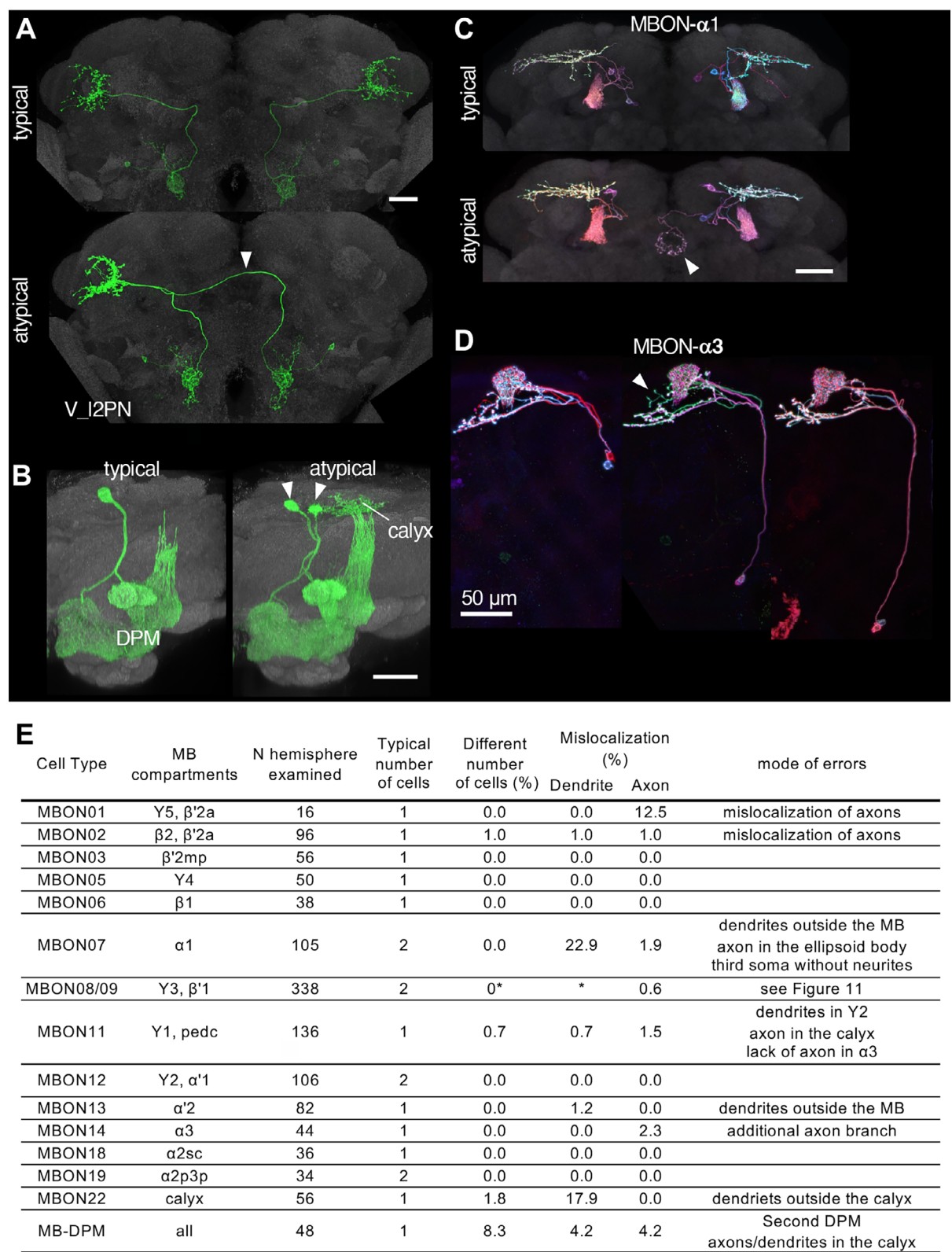

**Figure 10.** Stereotypy and erroneous projections. (**A**) V_l2PN from both hemispheres send axonal projections to the right hemisphere in an atypical case (arrowhead). (**B**) In this atypical case, there are two MB-DPM neurons in one hemisphere (arrowhead) and the arborizations extend beyond the pedunculus into the calyx. (**C**) MBON-α1 occasionally has additional arborizations in the ellipsoid body (arrowhead). (**D**) The localization of MBON-α3 soma widely varied along the dorso-ventral axis. It occasionally had an additional axonal branch (arrowhead). (**E**) A table to summarize normal and

Figure 10 continued

erroneous projections of MBONs and MB-DPM. In all the cases except for the DPM, 'different number of cells' was likely due to stochastic expression of the drivers (i.e. lack of labeling) rather than biological difference. We defined 'mislocalization' when axons or dendrites projected to outside of the normally targeted brain regions. For instance, dendrites of typical MBON07 are usually confined inside the α1, but were extended to outside the MB lobes in 22.9% of samples. Variable branching patterns inside the normally targeted brain regions were not counted as mislocalization here. In some MB310C-split-GAL4 samples, we observed a third soma attached to MBON-α1 but they lacked any neurites. We did not observe obvious mislocalization of dendrites or axons for MBON03, 5, 6, 12, 18, and 19. See **Figure 11** for variability of MBON08/09 in MB083C.

(**Supplementary file 1**). These lines, together with preceding collections of drivers (**Aso et al., 2014a**; **Aso and Rubin, 2016**; **Davis et al., 2020**; **Dolan et al., 2018**; **Rubin and Aso, 2024**; **Shuai et al., 2015**; see for examples: **Sterne et al., 2021**; **Strother et al., 2017**; **Truman et al., 2023**; **Tuthill et al., 2013**; **Wang et al., 2021**; **Wolff and Rubin, 2018**; **Wu et al., 2016**), collectively constitute a powerful resource for precise functional interrogation of associative learning in adult *Drosophila melanogaster*, and will be a foundation to reveal conserved principles of neural circuits for associative learning.

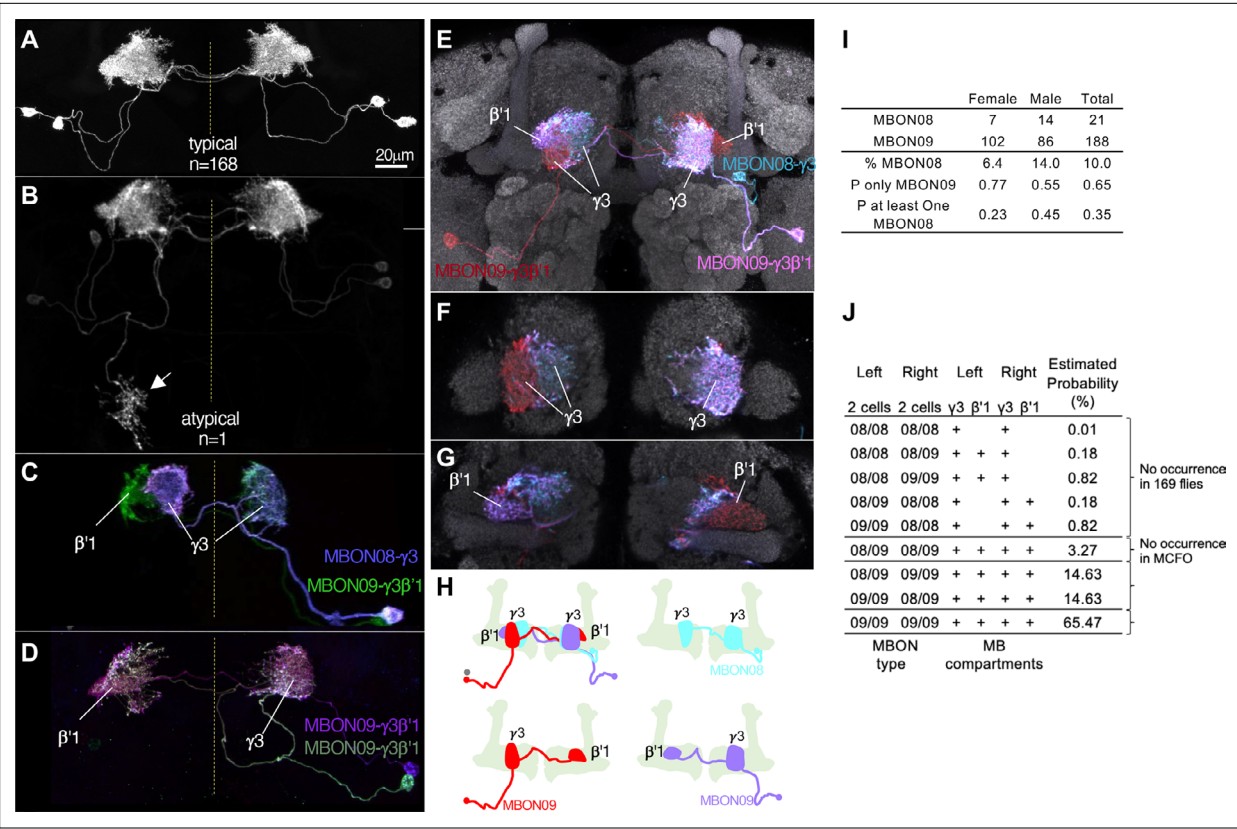

**Figure 11.** Individuality and asymmetry of MBON08 and MBON09. (**A**) A typical image of two MBONs in MB083C-split-GAL4 driver. (**B**) Abnormal axonal projection of MBON08/09 observed in one of 169 samples. (**C, D**) MCFO images of MB083C driver from different flies show that the two cells can either be both MBON09-γ3β'1 (**C**) or one MBON09-γ3β'1 and one MBON08-γ3 (**D**). (**E–G**) An example of MCFO image of MB083C, which visualized one MBON08 and two MBON09 in the same brain. The projection (**E**) and image sections at the γ3 (**F**) or β'1 (**G**) are shown. (**H**) Diagrams of the three MBONs shown in E-G. (**I**) A summary table for observation of MBON08 and MBON09 in male and female brains. (**J**) All possible variations of 4 MBONs in MB083C driver, and estimated probability for each case based on the observations summarized in I. (**A**) and (**C**) were adapted from Figure 8 of **Aso et al., 2014a**.

The online version of this article includes the following figure supplement(s) for figure 11:

**Figure supplement 1.** MB083C invariantly label two MBONs.

**Figure supplement 2.** Subdivisions of medial and lateral γ3 compartments.

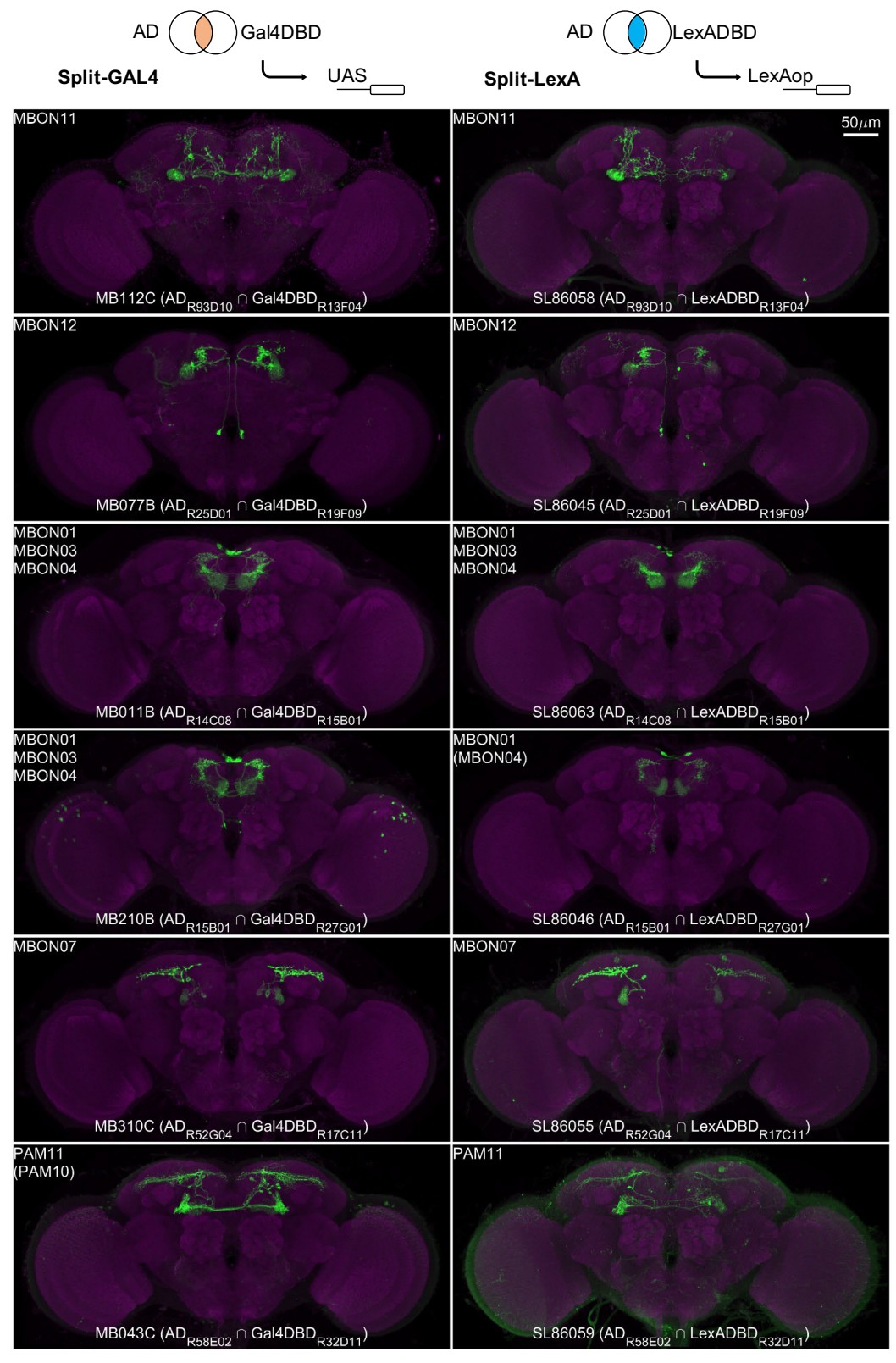

**Figure 12.** Examples of split-LexA conversion. Split-LexA shares the same enhancers with split-GAL4 but with the Gal4DBD replaced by LexADBD. Among 34 conversions tested, 22 were successful, with the split-LexA showing identical or similar expression patterns as the split-GAL4. The remaining 12 had weak/no expression or showed unintended intersectional patterns. See *Supplementary file 2* for the hemidriver lines with p10 translational enhancers to enhance expression level.

## Materials and methods

### Fly strains

*Drosophila melanogaster* strains were reared at 22 °C and 60% humidity on standard cornmeal food in 12:12 hour light:dark cycle. The genotypes of all split-GAL4 and split-LexA driver lines released here are listed in the *Supplementary file 1*. The new collection of split-GAL4 drivers reported here was designed based on confocal image databases (https://flweb.janelia.org/cgi-bin/flew.cgi) (*Jenett et al., 2012*; *Tirian and Dickson, 2017*), and screening expression patterns of p65ADZp and ZpGAL4DBD combinations was performed as described previously (*Aso et al., 2014a*; *Pfeiffer et al., 2010*). The confocal microscopy images of the splitGAL4 lines can be found at flylight database under '*Scaplen et al., 2021*' release. Fly strains can be requested from Janelia or the Bloomington stock center. Split-LexA expression data and fly strains are also available upon request from Aso lab.

### Immunohistochemistry

Brains and ventral nerve cords of 3–10 days old flies were dissected, fixed and immunolabeled and imaged with confocal microscopes (Zeiss LSM710, LSM780, or LSM880) as previously described (*Aso et al., 2014a*; *Jenett et al., 2012*; *Meissner et al., 2023*; *Nern et al., 2015*). The detailed protocols and videos are available at https://www.janelia.org/project-team/flylight/protocols.

Most samples were collected from females, though typically at least one male fly was examined for each driver line. While we noticed certain lines such as SS48900, exhibited distinct expression patterns in females and males, we did not particularly focus on sexual dimorphism, which is analyzed elsewhere (*Meissner et al., 2024*). Therefore, unless stated otherwise, the presented samples are of mixed gender. Detailed metadata, including gender information and the reporter used, can be found in *Supplementary file 7*.

### Whole-body sections

For sample preparation, flies were anesthetized on ice and briefly washed with 70% ethanol. Small incisions were made in the flanks of the thorax and abdomen under 2% paraformaldehyde in PBS with 0.1% Triton X-100 (PBS-T), and the flies were fixed in this solution overnight at 4 °C. After washing in PBS containing 1% Triton X-100, the samples were embedded in 7% agarose and sectioned on Leica Vibratome (VT1000s) sagittally in slices of 0.3 mm. The slices were incubated in PBS with 1% Triton X-100, 0.5% DMSO, 3% normal goat serum, Texas Red-X Phalloidin (1:50, Life Technologies #T7471) and anti-GFP rabbit polyclonal antibodies (1:1000, Thermo Fisher, #A10262) at room temperature with agitation for 24 hours. After a series of three washes in PBS-T, the sections were incubated for another 24 hr in the solution containing secondary antibodies (1:1000, goat anti-rabbit, Thermo Fisher #A32731). The samples were then washed in PBS-T and mounted in Tris-HCL (pH 8.0)-buffered 80% glycerol + 0.5% DMSO. For imaging and rendering, serial optical sections were obtained at 2 μm intervals on a Zeiss 880 confocal microscope with a pan-apochromat 10 x/0.45 NA objective using 488 and 594 nm lasers. Images were processed in Fiji (https://fiji.sc/) and Photoshop (Adobe Systems Inc).

### Behavioral assays

For flies expressing CsChrimson (*Klapoetke et al., 2014*), the food was supplemented with retinal (0.2 mM all-trans-retinal prior to eclosion and then 0.4 mM). Two- to 6-day-old adult females were collected and sorted on a Peltier cold plate 2–4 days before testing in behavioral assays. Flies were starved for 40–48 hr on 1% agar before they were subjected to behavioral experiments. Olfactory conditioning and optogenetic activation experiments were performed as previously described using the modified four-field olfactory arena equipped with the 627 nm LED board and odor mixers (*Aso and Rubin, 2016*; *Pettersson, 1970*). The odors were diluted in paraffin oil: pentyl acetate (PA, 1:10000, v/v) and ethyl lactate (EL, 1:10000, v/v). Videos were taken at 30 frames per second and analyzed using Fiji and Caltech FlyTracker (*Eyjolfsdottir et al., 2014*).

### LM-EM matching

The confocal microscopy images of different split-GAL4 lines were registered to a common template JRC2018_unisex (*Bogovic et al., 2020*) and organized in Janelia Workstation software (https://github.com/JaneliaSciComp/workstation; *JaneliaSciComp, 2025*). Color depth MIP mask search (*Otsuna*

et al., 2018) was used to search through the EM neuron library (hemibrain 1.2.1) for matching candidates. The searching target was obtained by either creating a mask on the full confocal image or using neurons of interest manually segmented in VVD viewer (https://github.com/takashi310/VVD_Viewer; Kawase et al., 2023; Wan et al., 2012). SWC files of top-matching EM neuron candidates were loaded into VVD viewer together with the confocal microscopy images in the same standard brain coordinates. By rotating the 3d images and manually scrutinizing the branching patterns, we picked the best matching candidate. Typically, we had high confidence of LM-to-EM matching for the line that labels only one cell per hemishere. For instance, we could unambiguously match the cell in SS67721 with SMP108 in the EM hemibrain volume. Our confidence of LM-to-EM matching tended to be lower for the lines that label multiple cells, because neurons of similar morphologies could be labeled together in those lines.

## Connectivity analysis

Connectivity information was retrieved from neuPrint (https://neuprint.janelia.org/), a publicly accessible website hosting the 'hemibrain' dataset (Scheffer et al., 2020). For cell types, we used cell type name assignments reported in Scheffer et al., 2020. Only connections of the cells in the right hemisphere were used due to incomplete connectivity in the left hemisphere. The 3D renderings of neurons presented were generated using the visualization tools of NeuTu (Zhao et al., 2018) or VVD viewer.

## Statistics

Statistical comparisons were performed on GraphPad Prism 7.0 using one-way ANOVA followed by Dunnett's test for multiple comparisons. Sample size was not predetermined based on pilot experiments.

## Detailed fly genotypes used by figures

| | |
|---|---|
| *Figure 1F* | w/w, 20xUAS-CsChrimson-mVenus attP18; +/split-GAL4<br>w/w, 20xUAS-CsChrimson-mVenus attP18; +/P{Gr64f-GAL4.9.7}5; +/P{Gr64f-GAL4.9.7}1 |
| *Figure 2A* | w/w, 20xUAS-CsChrimson-mVenus attP18;+/SS45245-*split-GAL4* |
| *Figure 2B* | w/w, pBPhsFlp2::PEST in attP3;;<br>pJFRC201-10XUAS-FRT>STOP > FRT-myr::smGFP-HA in VK00005,<br>pJFRC240-10XUAS-FRT>STOP > FRT-myr::smGFP-V5-THS-10XUAS-<br>FRT>STOP > FRT-myr::smGFP-FLAG in su(Hw)attP1/SS45245-*split-GAL4* |
| *Figure 2C* | w/w;pJFRC225-5XUAS-IVS-myr::smFLAG in VK00005, pJFRC51-3XUAS-IVS-Syt::smHA in su(Hw)attP1/SS45245-*split-GAL4* |
| *Figure 7E–J* | w/w, 20xUAS-CsChrimson-mVenus attP18;+/SS87269-*split-GAL4*<br>w/w, 20xUAS-CsChrimson-mVenus attP18;+/SS87272-*split-GAL4*<br>w/w, 20xUAS-CsChrimson-mVenus attP18;+/SS88801-*split-GAL4*<br>w/w, 20xUAS-CsChrimson-mVenus attP18;+/SS87278-*split-GAL4* |
| *Figure 7K–Q* | w/w, 20xUAS-CsChrimson-mVenus attP18;+/Gr64f-split-GAL4s w/w,<br>20xUAS-CsChrimson-mVenus attP18;+/Gr66a-GAL4 |
| *Figure 7—figure supplement 1* | w/w; +/ P{Gr64f-GAL4.9.7}5; 5xUAS-myr-smFLAG in VK00005/P{Gr64f-GAL4.9.7}1 |
| *Figures 2* | w/w, 20xUAS-CsChrimson-mVenus attP18; +/Gr64f-split-GAL4s w/w,<br>20xUAS-CsChrimson-mVenus attP18; +/ P{Gr64f-GAL4.9.7}5; +/P{Gr64f-GAL4.9.7}1<br>w/w, 20xUAS-CsChrimson-mVenus attP18; +/empty-split-GAL4 |
| *Figure 8—figure supplement 1A* | w/w, 20xUAS-CsChrimson-mVenus attP18; +/SS87269-*split-GAL4* |
| *Figure 8—figure supplement 1C* | w/w, 20xUAS-CsChrimson-mVenus attP18; +/SS87269-*split-GAL4*<br>w/w, 20xUAS-CsChrimson-mVenus attP18; +/SS87278-*split-GAL4*<br>w/w, 20xUAS-CsChrimson-mVenus attP18; +/empty-*split-GAL4* |

*Continued on next page*

*Continued*

| | |
|---|---|
| *Figure 8I* | *w/w, 20xUAS-CsChrimson-mVenus attP18;+/SS87269-split-GAL4*<br>*w/w, 20xUAS-CsChrimson-mVenus attP18;+/SS88801-split-GAL4*<br>*w/w, 20xUAS-CsChrimson-mVenus attP18;+/empty-GAL4* |
| *Figure 9B–D* | *w/w, 20xUAS-CsChrimson-mVenus attP18;+/SS01159-split-GAL4*<br>*w/w, 20xUAS-CsChrimson-mVenus attP18;+/empty-GAL4* |
| *Figure 10A* | *w/w;pJFRC225-5XUAS-IVS-myr::smFLAG in VK00005, pJFRC51-3XUAS-IVS-Syt::smHA in su(Hw)attP1/SS01336-split-GAL4* |
| *Figure 10B* | *w/w;pJFRC225-5XUAS-IVS-myr::smFLAG in VK00005, pJFRC51-3XUAS-IVS-Syt::smHA in su(Hw)attP1 /SS01241-split-GAL4* |
| *Figure 10C* | *w/w, pBPhsFlp2::PEST in attP3;; pJFRC201-10XUAS-FRT>STOP > FRT-myr::smGFP-HA in VK00005, pJFRC240-10XUAS-FRT>STOP > FRT-myr::smGFP-V5-THS-10XUAS-FRT>STOP > FRT-myr::smGFP-FLAG in su(Hw)attP1/MB310C-split-GAL4* |
| *Figure 10D* | *w/w, pBPhsFlp2::PEST in attP3;; pJFRC201-10XUAS-FRT>STOP > FRT-myr::smGFP-HA in VK00005, pJFRC240-10XUAS-FRT>STOP > FRT-myr::smGFP-V5-THS-10XUAS-FRT>STOP > FRT-myr::smGFP-FLAG in su(Hw)attP1/MB082C-split-GAL4* |
| *Figures 1* | *w/w;;pJFRC225-5XUAS-IVS-myr::smFLAG in VK00005, pJFRC51-3XUAS-IVS-Syt::smHA in su(Hw)attP1 /MB083C-split-GAL4*<br>*w/w;UAS-mCD8::GFP/MB083C-split-GAL4*<br>*w/w, 20xUAS-CsChrimson-mVenus in attP18;;+/ MB083C-split-GAL4*<br>*w/w, pJFRC12-10XUAS-IVS-myr::GFP in attP18 /MB083C-split-GAL4* |
| *Figure 11—figure supplement 2* | *w, 10xUAS-IVS-myr::smGdP-HA in attP18, 13xLexAop2-IVS-myr::smGdP-V5 in su(Hw)attP8; +/R52G04-LexA (MBON08/09);+/MB312C-split-GAL4 (PAM-γ4)*<br>*w, 10xUAS-IVS-myr::smGdP-HA in attP18, 13xLexAop2-IVS-myr::smGdP-V5 in su(Hw)attP8; +/R52G04-LexA (MBON08/09);+/MB441B-split-GAL4 (PAM-γ3) w/w;;VT006202-LexAp65 in attP2/pJFRC19-13XLexAop2-IVS-myr::GFP in attP2* |
| *Figure 12* | *w/w, 20xUAS-CsChrimson-mVenus attP18; +/split-GAL4s w/w, 13xLexAop2-CsChrimson-mVenus attp18; +/split-LexAs* |

## Acknowledgements

We thank Toshihide Hige, Daisuke Hattori, members of the YA, GMR and GT laboratories for valuable comments on the manuscript. We thank all the members of Janelia Flylight and Project Technical Resources for technical assistance for constructing split-GAL4 drivers and generating confocal microscopy images. During this effort, the FlyLight Project Team and Project Technical Resources included Gudrun Ihrke, Megan Atkins, Shelby Bowers, Kari Close, Gina DePasquale, Zack Dorman, Kaitlyn Forster, Jaye Anne Gallagher, Theresa Gibney, Asish Gulati, Joanna Hausenfluck, Yisheng He, Kristin Hendersen, Hsing Hsi Li, Nirmala Iyer, Jennifer Jeter, Lauren Johnson, Rebecca Johnston, Rachel Lazarus, Kelley Lee, Hua-Peng Liaw, Oz Malkesman, Geoffrey Meissner, Brian Melton, Scott Miller, Reeham Motaher, Alexandra Novak, Omatara Ogundeyi, Alyson Petruncio, Jacquelyn Price, Sophia Protopapas, Susana Tae, Athreya Tata, Jennifer Taylor, Allison Vannan, Rebecca Vorimo, Brianna Yarborough, Kevin Xiankun Zeng, and Chris Zugates, with Steering Committee of YA, GMR, Gwyneth Card, Barry Dickson, Reed George, Wyatt Korff, and James Truman. We also thank Kelly Ashley, Pria Chang, Tam Dang, Dona Fetter, Guillermo Gonzalez, Donald Hall, Jui-Chun Kao, James McMahon, Monti Mercer, Brenda Perez, Scarlett Pitts, Danielle Ruiz, Brandi Sharp, Viruthika Vallanadu, Grace Zheng, Amanda Cavallaro, Todd Laverty of Janelia Fly Facility for husbandry of stocks, and Eric Trautman, Rob Svirskas, Hideo Otsuna, Takashi Kawase and other members of Janelia Scientific Computing for supporting organization and analysis of confocal and EM microscopy images.

# Additional information

## Funding

| Funder | Grant reference number | Author |
|--------|------------------------|--------|
| Howard Hughes Medical Institute | | Tzumin Lee<br>Gerald M Rubin<br>Glenn C Turner<br>Yoshinori Aso |

The funders had no role in study design, data collection and interpretation, or the decision to submit the work for publication.

## Author contributions

Yichun Shuai, Conceptualization, Data curation, Formal analysis, Validation, Investigation, Visualization, Writing – original draft, Writing – review and editing; Megan Sammons, Data curation, Formal analysis, Investigation; Gabriella R Sterne, Resources, Formal analysis, Validation, Investigation, Methodology, Writing – original draft; Karen L Hibbard, Resources, Methodology; He Yang, Investigation; Ching-Po Yang, Claire Managan, Data curation; Igor Siwanowicz, Investigation, Methodology; Tzumin Lee, Supervision, Writing – review and editing; Gerald M Rubin, Glenn C Turner, Conceptualization, Supervision, Funding acquisition, Writing – original draft, Project administration, Writing – review and editing; Yoshinori Aso, Conceptualization, Resources, Data curation, Formal analysis, Supervision, Funding acquisition, Investigation, Visualization, Writing – original draft, Project administration, Writing – review and editing

## Author ORCIDs

Yichun Shuai  https://orcid.org/0000-0001-9243-425X
Karen L Hibbard  https://orcid.org/0000-0002-2001-6099
Igor Siwanowicz  https://orcid.org/0000-0001-5819-1530
Tzumin Lee  https://orcid.org/0000-0003-0569-0111
Gerald M Rubin  https://orcid.org/0000-0001-8762-8703
Glenn C Turner  https://orcid.org/0000-0002-5341-2784
Yoshinori Aso  https://orcid.org/0000-0002-2939-1688

Reviewer #1 (Public Review): https://doi.org/10.7554/eLife.94168.4.sa1
Reviewer #2 (Public Review): https://doi.org/10.7554/eLife.94168.4.sa2
Reviewer #3 (Public Review): https://doi.org/10.7554/eLife.94168.4.sa3
Author response https://doi.org/10.7554/eLife.94168.4.sa4

# Additional files

## Supplementary files

Supplementary file 1. A list of released driver lines and their corresponding EM neurons. Listed are stable split-GAL4 lines (SS, MB) and split-LexA lines (SL). Drivers are grouped by the anatomical regions where they have their primary expression. Matching to EM cell types in the hemibrain v1.2.1 dataset was performed by an algorithmic search of the morphologies of EM skeletons, followed by manual evaluation of proposed matches using 3D visualization software. The best-matching candidates are listed. Where multiple candidates seem valid, the alternatives are listed in the note column. In some cases, a matching EM BodyId can be found in the hemibrain v1.2.1 dataset but an EM cell type has not been defined; these are listed as TBD. Number of cells observed in light microscopy (LM) or electron microscopy (EM) are from one hemisphere, unless otherwise stated. A descriptive cell type column is included, which uses conventional names (if available) to facilitate searching and includes cell types with weak/stochastic expression (bracketed) or putative cell types with less matching confidence. When there are multiple drivers for the same cell type(s), one is chosen as the primary driver based on specificity, consistency, and expression strength. Abbreviations: AL, antennal lobe; ORNs, olfactory receptor neurons; AL LNs, antennal lobe local interneurons; PNs, projection neurons; MB, mushroom body; KCs, Kenyon cells; MBONs, mushroom body output neurons; DANs, dopaminergic neurons; LH, lateral horn; LHONs, lateral horn output

neurons; LHLNs, lateral horn local neurons; SNP, superior neuropils; CRE, crepine; CX, central complex; LAL, lateral accessory lobe; ANs, ascending neurons; DNs, descending neurons; OANs, octopaminergic neurons; 5-HT, serotonergic neurons; OL, optic lobe; VLNP, ventrolateral neuropils; INP, inferior neuropils (other than crepine); VMNP, ventromedial neuropils; PENP, periesophageal neuropils; GNG, gnathal ganglia; VNC, ventral nerve cord; N.D., not determined.

Supplementary file 2. New transgenic flies generated in this study. The enhancer fragments, insertion sites, and inserted chromosomes used to construct the lines are listed. For some of the transgenes, an additional version with a *p10* 3'-UTR (*Pfeiffer et al., 2012*) was generated to increase the expression.

Supplementary file 3. Coverage of MBON-downstream and DAN-upstream. Connection matrix between MB interneurons and DANs and MBONs. A threshold was set to exclude connections with a low number of neuron-neuron connections, specifically, 10 connections for MBON to a downstream neuron and 5 connections for upstream neurons to a DAN (*Li et al., 2020*). Recurrent neurons are defined as cell types receiving input from MBONs and supplying output to DANs. Neurotransmitter (NT) prediction data were from *Eckstein et al., 2023*, and the fraction of synapses predicted for the neurotransmitter was pooled from all cells of the cell type.

Supplementary file 4. List of non-KC cell types within the MB. The list came from a query in hemibrain 1.2.1 for neurons that have either ≥ 50 pre-synaptic connections or ≥ 250 post-synaptic connections in the MB region of interest (excluding the accessory calyx) on the right hemisphere. KCs were intentionally omitted from the list. Other than PNs, MBONs and DANs, the list also highlights a couple of cell types interconnecting LH and MB.

Supplementary file 5. Updated list of driver lines for cell types within the MB excluding Kcs. This includes new or improved split-GAL4 and split-LexA lines from the present study, lines from the *Aso et al., 2014a* collection (*Aso et al., 2014a*), a recent MBON collection (*Rubin and Aso, 2024*), MB630B (*Aso and Rubin, 2016*), SS01308 (*Aso et al., 2019*), MB063B, SS23107 and SS23112 (*Dolan et al., 2019*), SS46348 (*Otto et al., 2020*), and some regular Gal4 lines VT43924-Gal4.2 (*Amin et al., 2020*) and G0239 (*Chiang et al., 2011*). Lines listed in boldface are generally of higher quality.

Supplementary file 6. Coverage of PN cell types. A list of split-GAL4 lines and their coverage of PNs of the antennal lobe. Shading indicates expression level. Many of the multi-glomerular PN (mPN) cell types cannot be easily differentiated based on light microscopy images, so they are listed as a broad mPN category in the table.

Supplementary file 7. Metadata for presented samples. Detailed metadata organized by figures, including information about genotype, gender and the reporter used. Similar metadata can be found on FlyLight website for deposited confocal imaging data.

MDAR checklist

## Data availability

The confocal images of expression patterns are available at https://splitgal4.janelia.org/cgi-bin/splitgal4.cgi. The values used for figures are summarized in source data.

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

# Appendix 1

## Appendix 1—key resources table

| Reagent type (species) or resource | Designation | Source or reference | Identifiers | Additional information |
|---|---|---|---|---|
| Strain, strain background (*Drosophila melanogaster*) | New split-GAL4 and split-LexA lines | This paper; https://splitgal4.janelia.org/cgi-bin/splitgal4.cgi | | Available from Aso lab |
| Strain, strain background (*D. melanogaster*) | 20xUAS-CsChrimson- mVenus attP18 | *Klapoetke et al., 2014*; PMID:24509633 | | |
| Strain, strain background (*D. melanogaster*) | pJFRC200-10xUAS- IVS-myr::smGFP-HA in attP18 | *Nern et al., 2015*; PMID:25964354 | | |
| Strain, strain background (*D. melanogaster*) | pJFRC225-5xUAS- IVS-myr::smGFP-FLAG in VK00005 | *Nern et al., 2015*; PMID:25964354 | | |
| Strain, strain background (*D. melanogaster*) | pBPhsFlp2::PEST in attP3 | *Nern et al., 2015*; PMID:25964354 | | |
| Strain, strain background (*D. melanogaster*) | pJFRC201-10XUAS-FRT>STOP > FRT-myr::smGFP-HA in VK0005 | *Nern et al., 2015*; PMID:25964354 | | |
| Strain, strain background (*D. melanogaster*) | pJFRC240-10XUAS-FRT>STOP > FRT-myr::smGFP-V5-THS-10XUAS-FRT>STOP > FRT-myr::smGFP-FLAG_in_su(Hw)attP1 | *Nern et al., 2015*; PMID:25964354 | | |
| Strain, strain background (*D. melanogaster*) | empty-split-GAL4 (p65ADZp attP40, ZpGAL4DBD attP2) | *Hampel et al., 2015*; PMID:26344548 | RRID:BDSC_79603 | |
| Strain, strain background (*D. melanogaster*) | empty-Gal4 (pBDPGal4U attP2) | *Pfeiffer et al., 2008*; PMID:18621688 | RRID:BDSC_68384 | |
| Strain, strain background (*D. melanogaster*) | MB083C split-GAL4 | *Aso et al., 2014a*; PMID:25535793 | RRID:BDSC_68287 | Available from Aso lab |
| Strain, strain background (*D. melanogaster*) | w*; P{Gr64f-GAL4.9.7}5/CyO; P{Gr64f-GAL4.9.7}1/TM3, Sb[1] | *Haberkern et al., 2019*; PMID:31056392 | RRID:BDSC_57668 RRID:BDSC_57669 | |
| Strain, strain background (*D. melanogaster*) | Gr66a-Gal4 | *Joseph and Heberlein, 2012*; PMID:22798487 | | |
| Antibody | anti-GFP (rabbit polyclonal) | Invitrogen | A11122; RRID:AB_221569 | 1:1000 |
| Antibody | anti-Brp (mouse monoclonal) | Developmental Studies Hybridoma Bank | nc82; RRID:AB_2341866 | 1:30 |
| Antibody | anti-HA-Tag (mouse monoclonal) | Cell Signaling Technology | C29F4; #3724; RRID:AB_10693385 | 1:300 |
| Antibody | anti-FLAG (rat monoclonal) | Novus Biologicals | NBP1-06712; RRID:AB_1625981 | 1:200 |
| Antibody | anti-V5-TAG Dylight-549 (mouse monoclonal) | Bio-Rad | MCA2894D549GA; RRID:AB_10845946 | 1:500 |
| Antibody | anti-mouse IgG(H&L) AlexaFluor-568 (goat polyclonal) | Invitrogen | A11031; RRID:AB_144696 | 1:400 |
| Antibody | anti-rabbit IgG(H&L) AlexaFluor-488 (goat polyclonal) | Invitrogen | A11034; RRID:AB_2576217 | 1:800 |
| Antibody | anti-mouse IgG(H&L) AlexaFluor-488 conjugated (donkey polyclonal) | Jackson Immuno Research Labs | 715-545-151; RRID:AB_2341099 | 1:400 |
| Antibody | anti-rabbit IgG(H&L) AlexaFluor-594 (donkey polyclonal) | Jackson Immuno Research Labs | 711-585-152; RRID:AB_2340621 | 1:500 |
| Antibody | anti-rat IgG(H&L) AlexaFluor-647 (donkey polyclonal) | Jackson Immuno Research Labs | 712-605-153; RRID:AB_2340694 | 1:300 |
| Antibody | anti-mouse IgG(H&L) ATTO 647 N (goat polyclonal) | ROCKLAND | 610-156-121; RRID:AB_10894200 | 1:100 |

*Appendix 1 Continued on next page*

*Appendix 1 Continued*

| Reagent type (species) or resource | Designation | Source or reference | Identifiers | Additional information |
|---|---|---|---|---|
| Antibody | anti-rabbit IgG(H+L) Alexa Fluor 568 (goat polyclonal) | Invitrogen | A-11036; RRID:AB_10563566 | 1:1000 |
| Chemical compound, drug | Pentyl acetate | Sigma-Aldrich | 109584 | 1:10000 in paraffin oil |
| Chemical compound, drug | Ethyl lactate | Sigma-Aldrich | W244015 | 1:10000 in paraffin oil |
| Chemical compound, drug | Paraffin oil | Sigma-Aldrich | 18512 | |
| Software, algorithm | ImageJ and Fiji | NIH; https://imagej.nih.gov/ij/; *Schindelin et al., 2012*; http://fiji.sc/ | RRID:SCR_003070; RRID:SCR_002285 | |
| Software, algorithm | MATLAB | MathWorks; https://www.mathworks.com/ | RRID:SCR_001622 | |
| Software, algorithm | Adobe Illustrator CC | Adobe Systems; https://www.adobe.com/products/illustrator.html | RRID:SCR_010279 | |
| Software, algorithm | GraphPad Prism 9 | GraphPad Software; https://www.graphpad.com/scientific-software/prism/ | RRID:SCR_002798 | |
| Software, algorithm | Python | Python Software Foundation; https://www.python.org/ | RRID:SCR_008394 | |
| Software, algorithm | Caltech FlyTracker | *Eyjolfsdottir et al., 2014*; *Taylor and Branson, 2024*; https://github.com/kristinbranson/FlyTracker | | |
| Software, algorithm | Animal Part Tracker | *Kabra et al., 2022*; *Branson, 2024*; https://github.com/kristinbranson/APT | | |
| Software, algorithm | neuPrint | *Plaza et al., 2022*; https://neuprint.janelia.org/ | | |
| Software, algorithm | Cytoscape | *Shannon et al., 2003*; https://cytoscape.org/ | RRID:SCR_003032 | |
| Software, algorithm | Janelia workstation | HHMI Janelia; https://doi.org/10.25378/janelia.8182256.v1 | | |
| Software, algorithm | NeuTu | *Zhao et al., 2018*; *janelia-flyem, 2024*; https://github.com/janelia-flyem/NeuTu | | |
| Software, algorithm | VVD Viewer | *Wan et al., 2012*; *Kawase et al., 2023*; https://github.com/takashi310/VVD_Viewer | RRID:SCR_021708 | |
| Other | Grade 3 MM Chr Blotting Paper | Whatman | 3030–335 | Used in glass vials with paraffin-oil diluted odors |
| Other | mass flow controller | Alicat | MCW-200SCCM-D | Mass flow controller used for the olfactory arena |

