## [Editor Report · eLife Assessment]

This **important** collection of over 800 new cell type-specific driver lines will be an invaluable resource for researchers studying associative learning in *Drosophila*. Thoroughly characterized and well documented, this collection will permit researchers to selectively target neurons that deliver information to, or receive it from, the memory center of the fly brain called the Mushroom Body. Given the wealth of new drivers and the genetic access they provide to over 300 cell types, this **compelling** work will be of interest not only to researchers studying the mechanisms of associative learning but more generally to those dissecting sensorimotor circuits in the fly nervous system.

---

## [Referee Report · Reviewer #1 (Public Review)]

Summary:

The emergence of *Drosophila* EM connectomes has revealed numerous neurons within the associative learning circuit. However, these neurons are inaccessible for functional assessment or genetic manipulation in the absence of cell-type-specific drivers. Addressing this knowledge gap, Shuai et al. have screened over 4000 split-GAL4 drivers and correlated them with identified neuron types from the "Hemibrain" EM connectome by matching light microscopy images to neuronal shapes defined by EM. They successfully generated over 800 split-GAL4 drivers and 22 split-LexA drivers covering a substantial number of neuron types across layers of the mushroom body associative learning circuit. They provide new labeling tools for olfactory and non-olfactory sensory inputs to the mushroom body; interneurons connected with dopaminergic neurons and/or mushroom body output neurons; potential reinforcement sensory neurons; and expanded coverage of intrinsic mushroom body neurons. Furthermore, the authors have optimized the GR64f-GAL4 driver into a sugar sensory neuron-specific split-GAL4 driver and functionally validated it as providing a robust optogenetic substitute for sugar reward. Additionally, a driver for putative nociceptive ascending neurons, potentially serving as optogenetic negative reinforcement, is characterized by optogenetic avoidance behavior. The authors also use their very large dataset of neuronal anatomies, covering many example neurons from many brains, to identify neuron instances with atypical morphology. They find many examples of mushroom body neurons with altered neuronal numbers or mistargeting of dendrites or axons and estimate that 1-3% of neurons in each brain may have anatomic peculiarities or malformations. Significantly, the study systematically assesses the individualized existence of MBON08 for the first time. This neuron is a variant shape that sometimes occurs instead of one of two copies of MBON09, and this variation is more common than that in other neuronal classes: 75% of hemispheres have two MBON09's, and 25% have one MBON09 and one MBON08. These newly developed drivers not only expand the repertoire for genetic manipulation of mushroom body-related neurons but also empower researchers to investigate the functions of circuit motifs identified from the connectomes. The authors generously make these flies available to the public. In the foreseeable future, the tools generated in this study will allow important advances in the understanding of learning and memory in *Drosophila*.

Strengths:

(1) After decades of dedicated research on the mushroom body, a consensus has been established that the release of dopamine from DANs modulates the weights of connections between KCs and MBONs. This process updates the association between sensory information and behavioral responses. However, understanding how the unconditioned stimulus is conveyed from sensory neurons to DANs, and the interactions of MBON outputs with innate responses to sensory context remains less clear due to the developmental and anatomic diversity of MBONs and DANs. Additionally, the recurrent connections between MBONs and DANs are reported to be critical for learning. The characterization of split-GAL4 drivers for 30 major interneurons connected with DANs and/or MBONs in this study will significantly contribute to our understanding of recurrent connections in mushroom body function.

(2) Optogenetic substitutes for real unconditioned stimuli (such as sugar taste or electric shock) are sometimes easier to implement in behavioral assays due to the spatial and temporal specificity with which optogenetic activation can be induced. GR64f-GAL4 has been widely used in the field to activate sugar sensory neurons and mimic sugar reward. However, the authors demonstrate that GR64f-GAL4 drives expression in other neurons not necessary for sugar reward, and the potential activation of these neurons could introduce confounds into training, impairing training efficiency. To address this issue, the authors have elaborated on a series of intersectional drivers with GR64f-GAL4 to dissect subsets of labeled neurons. This approach successfully identified a more specific sugar sensory neuron driver, SS87269, which consistently exhibited optimal training performance and triggered ethologically relevant local searching behaviors. This newly characterized line could serve as an optimized optogenetic tool for sugar reward in future studies.

(3) MBON08 was first reported by Aso et al. 2014, exhibiting dendritic arborization into both ipsilateral and contralateral γ3 compartments. However, this neuron could not be identified in the previously published *Drosophila* brain connectomes. In the present study, the existence of MBON08 is confirmed, occurring in one hemisphere of 35% of imaged flies. In brains where MBON08 is present, its dendrite arborization disjointly shares contralateral γ3 compartments with MBON09. This remarkable phenotype potentially serves as a valuable resource for understanding the stochasticity of neurodevelopment and the molecular mechanisms underlying mushroom body lobe compartment formation.

---

## [Referee Report · Reviewer #2 (Public Review)]

Summary:

The article by Shuai et al. describes a comprehensive collection of over 800 split-GAL4 and split-LexA drivers, covering approximately 300 cell types in *Drosophila*, aimed at advancing the understanding of associative learning. The mushroom body (MB) in the insect brain is central to associative learning, with Kenyon cells (KCs) as primary intrinsic neurons and dopaminergic neurons (DANs) and MB output neurons (MBONs) forming compartmental zones for memory storage and behavior modulation. This study focuses on characterizing sensory input as well as direct upstream connections to the MB both anatomically and, to some extent, behaviorally. Genetic access to specific, sparsely expressed cell types is crucial for investigating the impact of single cells on computational and functional aspects within the circuitry. As such, this new and extensive collection significantly extends the range of targeted cell types related to the MB and will be an outstanding resource to elucidate MB-related processes in the future.

Strengths:

The work by Shuai et al. provides novel and essential resources to study MB-related processes and beyond. The resulting tools are publicly available and, together with the linked information, will be foundational for many future studies. The importance and impact of this tool development approach, along with previous ones, for the field cannot be overstated. One of many interesting aspects arises from the anatomical analysis of cell types that are less stereotypical across flies. These discoveries might open new avenues for future investigations into how such asymmetry and individuality arise from development and other factors, and how it impacts the computations performed by the circuitry that contains these elements.

---

## [Referee Report · Reviewer #3 (Public Review)]

Summary:

Previous research on the *Drosophila* mushroom body (MB) has made this structure the best-understood example of an associative memory center in the animal kingdom. This is in no small part due to the generation of cell-type specific driver lines that have allowed consistent and reproducible genetic access to many of the MB's component neurons. The manuscript by Shuai et al. now vastly extends the number of driver lines available to researchers interested in studying learning and memory circuits in the fly. It is an 800-plus collection of new cell-type specific drivers target neurons that either provide input (direct or indirect) to MB neurons or that receive output from them. Many of the new drivers target neurons in sensory pathways that convey conditioned and unconditioned stimuli to the MB. Most drivers are exquisitely selective, and researchers will benefit from the fact that whenever possible, the authors have identified the targeted cell types within the *Drosophila* connectome. Driver expression patterns are beautifully documented and are publicly available through the Janelia Research Campus's Flylight database where full imaging results can be accessed. Overall, the manuscript significantly augments the number of cell type-specific driver lines available to the Drosophila research community for investigating the cellular mechanisms underlying learning and memory in the fly. Many of the lines will also be useful in dissecting the function of the neural circuits that mediate sensorimotor circuits.

Strengths:

The manuscript represents a huge amount of careful work and leverages numerous important developments from the last several years. These include the thousands of recently generated split-Gal4 lines at Janelia and the computational tools for pairing them to make exquisitely specific targeting reagents. In addition, the manuscript takes full advantage of the recently released *Drosophila* connectomes. Driver expression patterns are beautifully illustrated side-by-side with corresponding skeletonized neurons reconstructed by EM. A comprehensive table of the new lines, their split-Gal4 components, their neuronal targets, and other valuable information will make this collection eminently useful to end-users. In addition to the anatomical characterization, the manuscript also illustrates the functional utility of the new lines in optogenetic experiments. In one example, the authors identify a specific subset of sugar reward neurons that robustly promotes associative learning.

---

## [Author Response]

The following is the authors’ response to the previous reviews.

Comments on revised version:Overall, I thought the authors addressed my comments well with the possible exception of what is actually new here. This was the most important thing that I thought should be included in the revision. Although the authors rewrote the paragraph describing the lines presented in the paper, I still can't tell exactly which ones haven't been previously published. Their revised paragraph says that 40 lines have been "previously used," but Supplemental Table 1 shows references for over 200 of the lines, which sounds more reasonable based on papers that have come out.

We have modified the text in line 112-120 as below.

“Supplementary File 1 lists 859 lines (including split-LexA) and their detailed information, such as genotype, expression specificity, matched EM cell type(s), and recommended driver for each cell type. A small subset of 47 lines from this collection have been previously used in studies (Aso et al., 2023; Dolan et al., 2019; Gao et al., 2019; Scaplen et al., 2021; Schretter et al., 2020; Takagi et al., 2017; Xie et al., 2021; Yamada et al., 2023).”

For 842 lines among the 859 lines listed in Supplementary File 1, this study is the primary citation for future papers for the following reason:

In 2021 December, we deposited the confocal images of new split-GAL4 lines at Janelia Flylight website (http://www.janelia.org/split-gal4) without a publication to describe annotation of expression patterns, and we already started sharing the lines without restrictions. In 2023 September, we released the preprint of this study at bioRxiv (doi: https://doi.org/10.1101/2023.09.15.557808). Up to this point, 47 lines have been used in other studies. In Supplementary File 1, 30 of them attribute the citation credit to both this study and other papers, because this 2023 preprint was cited as the primary citation in those papers. Similarly, the omni paper to summarize all the eWort of generating split-GAL4 lines by Janelia Flylight team (https://doi.org/10.7554/eLife.98405.1) cite many lines from this paper. However, since this summary paper did not provide additional information such as functional characterization by behavioral experiments, we did not include it in Supplementary File 1 to clarify that this study is the primary citation for these lines. The remaining 17 lines were published before 2021. We included them for the convenience of users, and we attributed the primary citation to the already published papers.

Also, in the revised paragraph they state that "All transgenic lines newly generated in this study are listed in Supplementary File 2" but that table lists only the 36 LexA hemidriver lines! Confusingly, this comment cites the same 8 references as are cited for the 40 line that they say were previously published. I am thus only more confused about how many previously uncharacterized lines are presented in this paper.

We modified the text as below to clarify that “new lines” indicate LexA or DBD lines but not new combination of already published AD and DBD lines. We removed the 8 citations, which were mistakenly placed in the previous manuscript.

“The newly generated LexA, Gal4DBD and LexADBD lines are listed in Supplementary File 2. “